

# Early Pleistocene large mammals from Maka'amitalu, Hadar, lower Awash Valley, Ethiopia

John Rowan[1], Ignacio A. Lazagabaster[2], Christopher J. Campisano[3], Faysal Bibi[2], René Bobe[4,5,6], Jean-Renaud Boisserie[7,8], Stephen R. Frost[9], Tomas Getachew[7,10], Christopher C. Gilbert[11,12], Margaret E. Lewis[13], Sahleselasie Melaku[10,14], Eric Scott[15,16], Antoine Souron[17], Lars Werdelin[18], William H. Kimbel[3] and Kaye E. Reed[3]

[1] Department of Anthropology, University at Albany, Albany, New York, United States
[2] Museum für Naturkunde, Berlin, Germany
[3] Institute of Human Origins, School of Human Evolution and Social Change, Arizona State University, Tempe, Arizona, United States
[4] Primate Models for Behavioural Evolution, Institute of Cognitive and Evolutionary Anthropology, School of Anthropology, University of Oxford, Oxford, United Kingdom
[5] Gorongosa National Park, Sofala, Mozambique
[6] Interdisciplinary Center for Archaeology and Evolution of Human Behavior (ICArEHB), Universidade do Algarve, Faro, Portugal
[7] Laboratoire Paléontologie Évolution Paléoécosystèmes Paléoprimatologie, Université de Poitiers, Poitiers, France
[8] Centre Français des Etudes Ethiopiennes (CNRS and Ministère des Affaires Etrangères, Ambassade de France, Ethiopia), Addis Ababa, Ethiopia
[9] Department of Anthropology, University of Oregon, Eugene, Oregon, United States
[10] Authority for Research and Conservation of Cultural Heritage, Addis Ababa, Ethiopia
[11] Department of Anthropology, City University of New York, Hunter College, New York, United States
[12] New York Consortium in Evolutionary Primatology (NYCEP), New York, United States
[13] Biology Program, School of Natural Sciences and Mathematics, Stockton University, Galloway, New Jersey, United States
[14] Paleoanthropology and Paleoenvironment Program, Addis Ababa University, Addis Ababa, Ethiopia
[15] Cogstone Resource Management Inc, Orange, California, United States
[16] Department of Biology, California State University, San Bernardino, San Bernardino, California, United States
[17] PACEA, Université Bordeaux, Bordeaux, France
[18] Department of Palaeobiology, Swedish Museum of Natural History, Stockholm, Sweden

Corresponding author
John Rowan, jrowan@albany.edu

## ABSTRACT

The Early Pleistocene was a critical time period in the evolution of eastern African mammal faunas, but fossil assemblages sampling this interval are poorly known from Ethiopia's Afar Depression. Field work by the Hadar Research Project in the Busidima Formation exposures (~2.7–0.8 Ma) of Hadar in the lower Awash Valley, resulted in the recovery of an early *Homo* maxilla (A.L. 666-1) with associated stone tools and fauna from the Maka'amitalu basin in the 1990s. These assemblages are dated to ~2.35 Ma by the Bouroukie Tuff 3 (BKT-3). Continued work by the Hadar Research Project over the last two decades has greatly expanded the faunal collection. Here, we provide a comprehensive account of the Maka'amitalu large mammals (Artiodactyla, Carnivora, Perissodactyla, Primates, and Proboscidea) and discuss their paleoecological and biochronological significance. The size of the Maka'amitalu

assemblage is small compared to those from the Hadar Formation (3.45–2.95 Ma) and Ledi-Geraru (2.8–2.6 Ma) but includes at least 20 taxa. Bovids, suids, and *Theropithecus* are common in terms of both species richness and abundance, whereas carnivorans, equids, and megaherbivores are rare. While the taxonomic composition of the Maka'amitalu fauna indicates significant species turnover from the Hadar Formation and Ledi-Geraru deposits, turnover seems to have occurred at a constant rate through time as taxonomic dissimilarity between adjacent fossil assemblages is strongly predicted by their age difference. A similar pattern characterizes functional ecological turnover, with only subtle changes in dietary proportions, body size proportions, and bovid abundances across the composite lower Awash sequence. Biochronological comparisons with other sites in eastern Africa suggest that the taxa recovered from the Maka'amitalu are broadly consistent with the reported age of the BKT-3 tuff. Considering the age of BKT-3 and biochronology, a range of 2.4–1.9 Ma is most likely for the faunal assemblage.

## INTRODUCTION

The Early Pleistocene was a critical time period in the evolution of eastern Africa's mammalian faunas. This period includes a significant faunal turnover driven by a species origination pulse (*Werdelin & Lewis, 2005*; *Bibi & Kiessling, 2015*) and an increase in community richness (*Fortelius et al., 2016*; *Du & Alemseged, 2018*) just after ~2 Ma. In terms of paleoecology, Early Pleistocene faunas of eastern Africa are dominated by grassland-dwelling herbivores (*Bobe & Behrensmeyer, 2004*; *Faith, Rowan & Du, 2019*) and stable isotope evidence from paleosol carbonates and leaf waxes document extensive $C_4$ biomass across much of the region (*Cerling et al., 2011*; *Uno et al., 2016*). The time period from ~2.5–1.5 Ma was also important in human evolution—it documents the initial diversification of the genera *Homo* and *Paranthropus*, an increase in the density of archaeological assemblages suggesting a more carnivorous diet for some hominin taxa (*Roche et al., 1999*; *Ferraro et al., 2013*), and a potential change in how hominins interacted with large mammal faunas and impacted ecosystem structure (*Werdelin & Lewis, 2013a*; *Faith et al., 2020*).

Much of our knowledge of the Early Pleistocene in eastern Africa derives from the well-studied Koobi Fora, Nachukui, and Shungura formations of the Turkana Basin, Kenya and Ethiopia (*e.g.*, *Boisserie et al., 2008*; *Fortelius et al., 2016*), and Olduvai Gorge, Tanzania (*e.g.*, *Leakey, 1965*; *Bibi et al., 2018*). Although the Afar Depression of Ethiopia has provided a rich Pliocene record of fossil hominins and faunas (*e.g.*, *Kimbel, Johanson & Rak, 1994*; *Reed, 2008*; *White et al., 2006*, *2009*), younger fossiliferous assemblages are poorly known from this region. Faunas of Early Pleistocene age have been reported from Gona (*Everett, 2010*; *Semaw et al., 2020*) and the Middle Awash (*Kalb et al., 1982*; *de Heinzelin et al., 1999*), but no comprehensive account of these assemblages has yet been
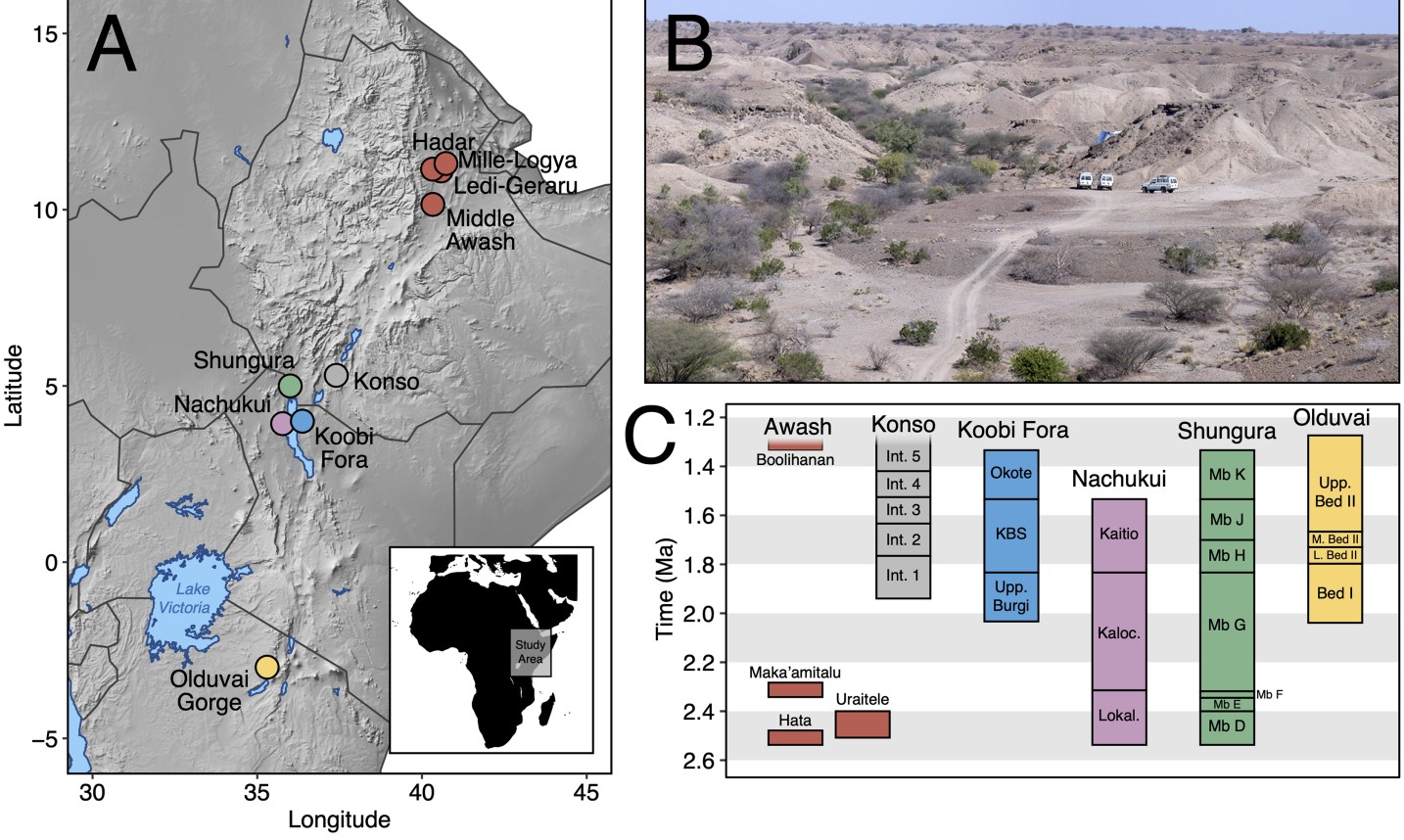

**Figure 1 The Maka'amitalu basin and other eastern African sites.** (A) Map of eastern Africa highlighting the major fossil-bearing sequences of the region. (B) Busidima Formation exposures at Hadar near localities A.L. 666 and A.L. 894 where hominin fossils and associated stone tools have been recovered. (C) Schematic summary of Early Pleistocene eastern African fossil sequences discussed in the text. 🖼 DOI: 10.7717/peerj.13210/fig-1

published. *Geraads et al. (2021)* recently provided a summary of the Mille-Logya fossil mammals, including the Early Pleistocene (~2.5–2.4 Ma) Uraitele exposures, but some taxa remain undescribed (*e.g.*, Cercopithecidae).

Here, we contribute new data on Early Pleistocene Afar faunas by providing a systematic paleontological account of the large mammals (orders Artiodactyla, Carnivora, Perissodactyla, Primates, and Proboscidea) from the Maka'amitalu basin in the Hadar Research Project area, lower Awash Valley, Ethiopia (Figs. 1A and 1B; Table 1). In addition to its fauna, the Maka'amitalu basin has yielded an early specimen of the genus *Homo* (maxilla A.L. 666-1) alongside Oldowan stone tool assemblages (*Kimbel et al., 1996*; *Goldman-Neuman & Hovers, 2009*; *Hovers, 2009*) dated to ~2.35 Ma (*Campisano & Feibel, 2008a*). Through our systematic description of the Maka'amitalu fauna, we provide taxonomic turnover and paleoecological analyses of faunal change in the lower Awash Valley between ~3.45–2.35 Ma and discuss the biochronological implications of the fauna in comparison with other Early Pleistocene fossil sites in eastern Africa (Fig. 1C).

**Table 1 Faunal list for large mammals from the Maka'amitalu basin of Hadar, Ethiopia.**

| Order | Family | Subfamily | Tribe | Taxon | NISP |
|---|---|---|---|---|---|
| Artiodactyla | Bovidae | Bovinae | Bovini | *Syncerus* sp. | 10 |
| | | | Tragelaphini | *Tragelaphus strepsiceros* | 19 |
| | | | | *Tragelaphus* sp. | 1 |
| | | Antilopinae | Alcelaphini | cf. *Beatragus antiquus* | 2 |
| | | | | *Parmularius altidens* or *P. angusticornis* | 1 |
| | | | | Alcelaphini gen. et. sp. indet. | 14 |
| | | | Antilopini | *Eudorcas praethomsoni* | 12 |
| | | | | Antilopini gen. et. sp. indet. | 1 |
| | | | Hippotragini | cf. *Oryx* sp. | 10 |
| | | | Reduncini | *Kobus sigmoidalis* or *K. ellipsiprymnus* | 5 |
| | | | | cf. *Redunca* or *Kobus kob* | 3 |
| | | | Indet. | Bovidae gen. et. sp. indet. | 1 |
| | Giraffidae | Giraffinae | | *Giraffa* sp. | 2 |
| | | Sivatheriinae | | *Sivatherium maurusium* | 2 |
| | Hippopotamidae | Hippopotaminae | | *Hippopotamus* cf. *gorgops* | 2 |
| | Suidae | Suinae | | *Kolpochoerus* cf. *phillipi* | 6 |
| | | | | *Metridiochoerus modestus* | 4 |
| Carnivora | Mustelidae | Lutrinae | | Lutrinae gen. et. sp. indet. | 1 |
| | Hyaenidae | | | Hyaenidae gen. et. sp. indet. | 1 |
| | Felidae | Felinae | | cf. *Acinonyx* sp. | 1 |
| | | | | Felidae gen. et. sp. indet. | 1 |
| | | Machairodontinae | Metailurini | *Dinofelis* cf. *aronoki* | 1 |
| Perissodactyla | Equidae | Equinae | | *Equus* sp. | 2 |
| Primates | Cercopithecidae | Cercopithecinae | Papionini | *Theropithecus oswaldi oswaldi* | 10 |
| | | | | Papionini gen. et. sp. indet. | 1 |
| | Hominidae | Homininae | Hominini | *Homo* sp. | 1 |
| | | | | Hominini gen. et. sp. indet. | 1 |
| Proboscidea | Elephantidae | | | *Elephas recki atavus* | 1 |

Note:
Faunal list based on specimens described here and previously described hominin remains by *Kimbel et al. (1996)*. NISP, number of identified specimens, based on unique specimen numbers (*e.g.*, A.L. 588-1A and A.L. 588-1B are considered a single specimen).

## Geological setting

The Hadar (~3.45 to 2.95 Ma) and Busidima (~2.7 to 0.8 Ma exposed) formations at the Hadar site (Figs. 2A and 2B) are separated by an angular unconformity above the spatially extensive BKT-2 marker tuff complex (~2.95 Ma), which caps the Hadar Formation (*Campisano & Feibel, 2008b*). This unconformity is thought to be the result of a period of rifting activity that ultimately formed a half-graben in which the Busidima Formation was subsequently deposited (*Quade et al., 2008*; *Wynn et al., 2008*). The Busidima Formation at Hadar consists of ~40 m of strata, which is much less extensive than in surrounding project areas, such as Dikika and Gona (*Campisano, 2012*). Similarly, the spatial extent of the Busidima Formation at Hadar covers only ~18 km$^2$ compared to Dikika and Gona where it encompasses ~100 km$^2$ and 200 km$^2$, respectively

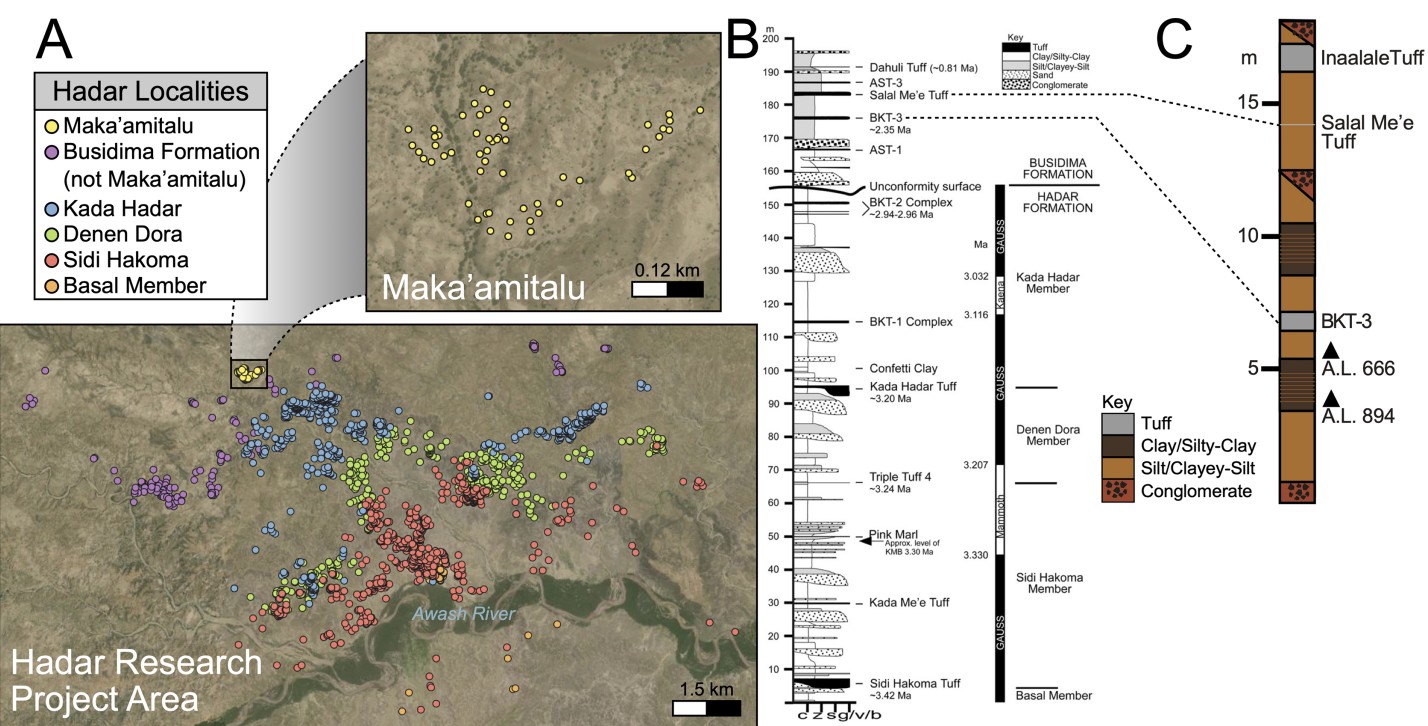

**Figure 2 Geologic summary of the Hadar Research Project area.** (A) Map of the Hadar Research Project area showing fossil localities colored by their respective geological member. (B) Composite stratigraphic section of the Hadar and Busidima formations at Hadar, adapted from *Campisano (2012)*. (C) Stratigraphic summary of the Maka'amitalu deposits. Key: c, clay; z = silt, s, sand; g/v/b, gravel/volcanic/bioclastic.

(*Campisano, 2012*; *Wynn et al., 2008*; *Quade et al., 2004*). The Busidima Formation at Hadar is dominated by erosional cut-and-fill fluvial sequences with a significantly reduced sedimentation rate (~2 cm/kyr at Hadar, ~5 cm/kyr at Gona) compared to the Hadar Formation (~30 cm/kyr) (*Campisano, 2012*; *Quade et al., 2008*).

There are 13 tuffs in the Busidima Formation that have been radiometrically dated, only two of which are exposed in the Hadar area (*Campisano, 2012*). At Hadar, Bouroukie Tuff 3 (BKT-3) is the most important marker tuff as it lies just above the A.L. 666 and A.L. 894 paleontological and archaeological localities (Figs. 2B and 2C). This tuff, traceable throughout the entire Maka'amitalu area as a tan to gray band varying between 35 to 80 cm in thickness, is located near the middle of the measured Maka'amitalu stratigraphic sequence (~7 m level in an ~18 m section; Fig. 2C) (*Campisano & Feibel, 2008a*). It was previously dated to 2.33 ± 0.07 Ma using plagioclase feldspars (*Kimbel et al., 1996*) and recalculated to 2.35 ± 0.07 Ma (*Campisano & Feibel, 2008a*) following recalibration of the Fish Canyon sanidine $^{40}$Ar/$^{39}$Ar standard. Unfortunately, subsequent attempts to re-date BKT-3 have not been as conclusive and the tephra has not been identified outside of the Hadar Research Project area, so potential correlations with neighboring sites cannot be made (*Campisano & Feibel, 2008a*). The Salal Me'e Tuff, located ~7 m above BKT-3, near the top of the Maka'amitalu sequence, is preliminarily dated to 2.2 Ma (*Campisano, 2007*) and may correlate with an unnamed tuff that *Walter et al. (1996)* dated to ~2 Ma.

All fossil specimens described in this work derive from the Maka'amitalu area at Hadar. The Maka'amitalu is a small basin of exposures and the specimens included in this study are all contained within a spatially restricted area of ~1 km² with a stratigraphic thickness of ~18 m (Figs. 2A, 2C). As with most of the Busidima Formation at Hadar, the depositional environment of the Maka'amitalu area is reconstructed as a high-energy fluvial system (Campisano, 2012). The Maka'amitalu sequence is dominated by overbank deposits of silts and clayey-silts without discrete fossiliferous horizons (Fig. 2C). Conglomerates within the Maka'amitalu represent fluvial channels, most of which are laterally discontinuous, but some, such as at the base of the section, can be traced across the basin. These conglomerates represent some degree of erosional unconformity, but no faults are noted. With the exception of material recovered *in situ* from excavations at the A.L. 666 and A.L. 894 localities, all faunal specimens were collected from surface exposures. The Authority for Research and Conservation of Cultural Heritage (ARCCH), Ethiopian Ministry of Culture and Tourism, and the Afar Regional State granted permission to conduct field work at Hadar.

# MATERIALS AND METHODS

## Analysis of fossil specimens

We examined all specimens of the orders Artiodactyla, Carnivora, Perissodactyla, Primates, and Proboscidea from the Maka'amitalu area in the Hadar Research Project collections housed in the National Museum of Ethiopia (Addis Ababa). These specimens were collected by the Hadar Research Project between 1993 and 2012. Our focus here is mainly on craniodental specimens because they are the most abundant and taxonomically informative, though given the small sample size of the Maka'amitalu assemblage, we include descriptions of postcrania when identifiable at or below the family level for Carnivora, Hippopotamidae, Equidae, and Primates.

Dental measurements are reported as mesiodistal lengths and buccolingual widths (MD × BL) for all taxa. Hypsodonty (unworn tooth crown height divided by width) was measured when possible. Basal dimensions of bovid horn cores are reported as anteroposterior and transverse (AP × TR) diameters. Horn core torsion, if present, is described as either homonymous or heteronymous: homonymous torsion is clockwise on the right side from the base up, whereas heteronymous torsion is anticlockwise on the right side from the base up (as in *Tragelaphus*). A complete list of specimens and measurements is provided in Data S1. All measurements were taken with digital calipers and are reported in millimeters (mm). Dental abbreviations used for systematic paleontological descriptions are as follows: Incisors are indicated by I or i, premolars by P or p, and molars by M or m, with uppercase letters referring to upper teeth and lowercase letters referring to mandibular teeth. Numbers 1, 2, 3, and 4 indicate tooth position (*e.g.*, M3 refers to the third upper molar).

## Turnover and paleoecological analyses

We compiled a presence-absence database of Hadar Formation, Ledi-Geraru, and Maka'amitalu herbivores (Artiodactyla, Perissodactyla, Proboscidea) to analyze turnover

and paleoecological change through time (Data S1). We focused our analyses on herbivores because they are abundant and most are large-bodied, meaning that they are unlikely to be impacted by the taphonomic and collection biases affecting the recovery of rare or small-bodied taxa (*Behrensmeyer, Kidwell & Gastaldo, 2000*). As primary consumers, they also directly reflect vegetation composition and other key ecosystem characteristics (*Greenacre & Vrba, 1984*; *Fortelius et al., 2016*). We divided the lower Awash sequence into submember and fault block units following *Campisano & Feibel (2008a)* and *DiMaggio et al. (2015)*, except for the adjacent Sidi Hakoma 4 (SH-4) and Denen Dora 1 (DD-1) submembers of the Hadar Formation; these units were analyzed as an aggregate (SH-4/DD-1) because they interdigitate across the Hadar outcrops and represent similar depositional environments.

**Turnover Analyses**—Patterns of taxonomic turnover between adjacent fossil units were quantified using the pairwise Simpson dissimilarity ($\beta_{sim}$) measure in the *betapart* package (*Baselga & Orme, 2012*) in R v.4.1 (*R Core Team, 2021*). $\beta_{sim}$ is calculated as

$$\beta_{sim} = \frac{\min(b, c)}{a + \min(b, c)}$$

where $a$ is the number of species common to both units, $b$ is the number of species that occur in the first unit but not in the second, and $c$ is the number of species that occur in the second unit but not the first. We chose $\beta_{sim}$ as our turnover measure because it quantifies species replacement independent of species richness (*Baselga, 2010*), which is variable in the lower Awash sequence. For turnover analyses we treated open nomenclature as follows: all records indeterminate at the species level were removed unless they were argued by taxonomic experts to represent distinct species (*e.g.*, *Kobus* sp. B of *Gentry (1981)*). We lumped 'cf.' species records with their probable species but retained 'aff.' records as distinct, as this is most often applied to specimens similar to but distinct from another taxon (*Bengtson, 1988*). Preliminary analyses comparing $\beta_{sim}$ values generated using our treatment of open nomenclature and the 'full' dataset (*i.e.*, retaining all indeterminate and 'cf.' records) were highly correlated (r = 0.97), suggesting that variation in taxonomic treatment has little impact on recovered turnover patterns.

**Paleoecological Analysis**—For paleoecological analysis, two categorical traits (body size and diet) were assigned to each herbivore species from the lower Awash sequence. Body size was recorded as a six-attribute system, s1 (<25 kg), s2 (25–75 kg), s3 (75–150 kg), s4 (150–350 kg), s5 (350–750 kg), s6 (>750 kg), primarily based on estimates from dentition or postcrania (*e.g.*, comparison of element sizes with extant species). Diet was based on a three-attribute system, $C_4$ grazer, $C_3$-$C_4$ mixed-feeder, and $C_3$ browser. Diet was assigned to species based on enamel stable carbon isotope data (*Robinson et al., 2017*; *Wynn et al., 2016*) and, in very few cases, on tribal or generic affiliation. We visualized paleoecological change through the lower Awash sequence by plotting diet and body size proportions through time. In addition, we compiled and plotted NISP (number of identified specimens) data for large-bodied bovid tribes (Aepycerotini, Alcelaphini, Antilopini, Bovini, Hippotragini, Reduncini, and Tragelaphini) from the Hadar Research Project and Ledi-Geraru Research

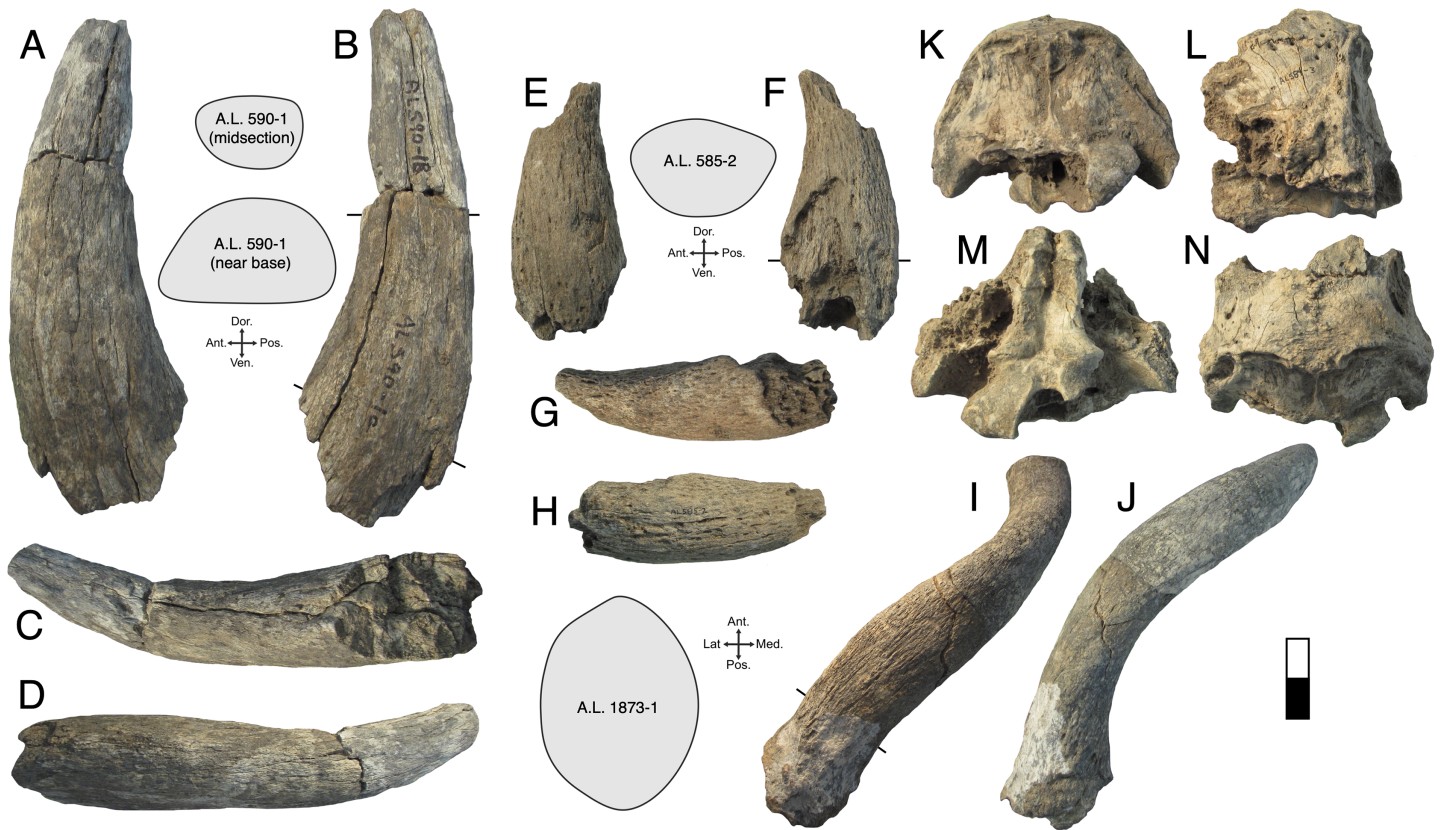

**Figure 3 Maka'amitalu Bovinae.** (A–D), A.L. 590-1 *Syncerus* sp. in ventral (A), dorsal (B), posterior (C), and anterior (D) view; (E–H), A.L. 585-2 *Syncerus* sp. in ventral (E), dorsal (F), posterior (G), and anterior (H) view. I–J, A.L. 1873-1 *Tragelaphus strepsiceros* in anterior (I) and lateral (J) view. K-N, A.L. 584-3 *T. strepsiceros* in posterior (K), lateral (L), ventral (M), and superior (N) view. Scale bar equals 5 cm.

Project databases. We quantified the evenness of tribal abundance using the Pielou index ($J'$), which is bound from 0 (composed of a single taxon) to 1 (complete evenness of taxa). The Pielou index is derived from the Shannon diversity index as $J' = H'/\ln(S)$ where $H'$ is the Shannon diversity index and $S$ the total number of species across all assemblages. The Shannon index is calculated as $H' = -\Sigma P_i(\ln P_i)$ where $P_i$ is the proportion of each taxon in the assemblage. We calculated evenness in PAST 4.06 (*Hammer, Harper & Ryan, 2001*) and estimated 95% confidence intervals with 10,000 bootstrap iterations.

**Systematic Paleontology**
Order Artiodactyla Owen, 1848
Family Bovidae Gray, 1821
Subfamily Bovinae Gray, 1821
Tribe Bovini Gray, 1821
Genus *Syncerus* Hogson, 1847
*Syncerus* sp.

**Referred specimens:** A.L. 590-1, horn core; A.L. 585-2, horn core; A.L. 586-11, left mandible with dp4-m2; A.L. 587-5, left M1/M2; A.L. 587-6, right m3 fragment; A.L. 589-4,

left M1/M2; A.L. 594-2, right M3; A.L. 598-2, left m3 fragment; A.L. 653-4, right P3; A.L. 658-1, right maxilla with M1-M3.

**Description:** A.L. 590-1 is a large bovin horn core broken near its base but comprising most of the mid-section (Figs. 3A–3D). As preserved, the basal dimensions are 98.5 × 57.5 mm, indicating that the horn core is strongly dorsoventrally (= transversely) compressed. The horn must have been relatively long (preserved length is ~320 mm), but it is difficult to tell exactly how close to the base the proximal break is because no sinuses are preserved. It is probably a left horn core, sweeping backward gently and lacking ventral dipping. Distally, the horn core curves upwards towards its tip. The basal horn core cross-section is roughly triangular. The ventral (= lateral) surface is very flat basally, with the dorsoventrally widest part of horn core located posteriorly. An anterior keel is present and seems to have been strong basally but weakens distally; all other keels are weak and rounded. The horn core cross-section becomes more oval distally, with the anterior and posterior dorsoventral depths more or less equal and keels rounded.

A.L. 585-2 is a smaller and much shorter left horn core (Figs. 3E–3H). The preserved length of A.L. 585-2 is ~175 mm, and this specimen shows a greater degree of distal tapering than the larger specimen; it was probably less than 300 mm long when complete. A.L. 585-2 shows evidence of strutted basal sinuses, so the proximal break must have been very near the base of the horn. Basal dimensions are 78 × 55e mm (the dorsal surface is eroded basally and the dorsoventral measurement had to be estimated). Its course is similar to that of the larger horn core A.L. 590-1: both are straight basally, with weak posterior curvature, and although the tip is not preserved on A.L. 585-2, the horn clearly curved upwards distally. A.L. 585-2 differs from A.L. 590-1 in that its posteromedial keel (= posterodorsal keel) is better developed than the anterior keel; there is no ventral flattening (as in the larger specimen) and instead the ventral side is much rounder; the widest anteroposterior part of the horn core occurs dorsally because of its rounded ventral surface. The anterior keel weakens distally whereas the posteromedial keel is perceptible at the distalmost section of what is preserved.

The Maka'amitalu horn cores differ from those of *Simatherium* and *Pelorovis* in their keels, compressed triangular cross-sections, and shorter lengths. *Ugandax coryndonae* from Hadar (*Gentry, 2006*) is a closer match, though the Maka'amitalu specimens differ in their greater compression, less triangular cross-sections, and being longer (at least in A.L. 590-1, which surpasses most complete *Ugandax* horns); all of these characters align the Maka'amitalu specimens with *Syncerus*. *Gentry & Gentry (1978)* named *Syncerus acoelotus* from the Early Pleistocene of Olduvai Gorge and recognized both a short- and long-horned form. A.L. 590-1 shares with the *S. acoelotus* holotype (Kar K II 1962.068/5811) moderately long horn cores that do not dip inferiorly, curve gently backward and upward, and have subtriangular cross-sections with a flattened anterior surface that becomes oval distally (*Gentry & Gentry, 1978*; plate 2). A.L. 582-2 is similar to the short-horned frontlet from Elephant K II above the Lemuta Member (*Gentry & Gentry, 1978*; plate 4, fig. 2) and shares short, squat horns with subtriangular cross-sections and a well-defined dorsal surface. However, the Maka'amitalu specimens differ from the long-horned Olduvai form in shorter length (horn length on the *S. acoelotus* holotype is ~650 mm and

A.L. 590-1 would have been clearly shorter when complete) and from the short-horned form in weaker curvature. In these respects, the Maka'amitalu horn cores recall the geologically older and less advanced specimens from the Shungura Formation that *Gentry (1985)* attributed to *Syncerus* '*?acoelotus*' (see *Bibi, Rowan & Reed, 2017* for a recent discussion of these specimens). Both Maka'amitalu horns differ from *S. caffer* in their more triangular cross-sections and lack of basal bossing, extensive hollowing of the basal horn core (the sinus on A.L. 585-2 is more similar to the moderate sinuses of *S. acoelotus*), and ventral dipping of the horn core course.

Isolated dentitions are identified as bovin based on their large size, large basal pillars, and complicated central cavities (*Gentry & Gentry, 1978*). They are presumed to belong to *Syncerus* as this is the only bovin identified in the Maka'amitalu assemblage on the basis of horn core material. They are smaller than those from a modern sample of the common cape buffalo *Syncerus caffer caffer* ($n = 5$). For example, M1 mesiodistal lengths are $22.4 \pm 3.18$ mm for the Maka'amitalu ($n = 3$) and $27.48 \pm 2.14$ mm for *S. c. caffer*, while M3 mesiodistal averages are $29.6 \pm 0.14$ mm for the Maka'amitalu specimens ($n = 2$) and $32.3 \pm 0.97$ mm for *S. c. caffer*. They are of similar size to teeth of *Syncerus* '*?acoelotus*' from Shungura Formation Members C and G (*Gentry, 1985*) but are smaller than teeth of *S. acoelotus* from Olduvai (*Gentry & Gentry, 1978*).

**Discussion:** *Syncerus* is probably first known from ~2.8 Ma sediments in Member C of the Shungura Formation (*Gentry, 1985*) and Ledi-Geraru (*Bibi, Rowan & Reed, 2017*) in Ethiopia, but this early species remains poorly known and unnamed. Younger in age is *Syncerus acoelotus*, best known from Olduvai Bed II, where it is represented by several relatively complete specimens (*Gentry & Gentry, 1978*). Given the fragmentary nature of the Maka'amitalu specimens and their similarities to both Olduvai *S. acoelotus* and older Shungura *S.* '*?acoelotus*', we refrain from attributing them to a species. The Omo (*Gentry, 1985*), Olduvai (*Gentry & Gentry, 1978*), and Daka (*Gilbert, 2008a*) *Syncerus* specimens are represented by both a short-horned and a long-horned morph. This could suggest that the *Syncerus* lineage may have always been polymorphic, as *S. caffer* is today (*Klein, 1994*).

Tribe Tragelaphini Blyth, 1863
Genus *Tragelaphus* de Blainville, 1816
*Tragelaphus strepsiceros* (Pallas, 1766)

**Referred specimens:** A.L. 584-3, occipital; A.L. 586-2, horn core; A.L. 589-1, horn core; A.L. 596-4, horn core; A.L. 1873-1, horn core. A.L. 586-3, right mandible with dp3-m1; A.L. 586-5, left m3; A.L. 586-8, left mandible with dp4-m1; A.L. 592-4, left mandible with p3; A.L. 593-3, left m3; A.L. 608-1, left mandible with p4-m3; A.L. 652-3, right mandible with m1-m2; A.L. 653-1, left maxilla with dP2-dP4; A.L. 659-2, right mandible with dp4; A.L. 666-11, left M1/M2; A.L. 666-12, left m1/m2; A.L. 666-21, left mandible with m1; A.L. 1872-1, right mandible with m1-m3; A.L. 1878-1, left mandible with m1-m2.

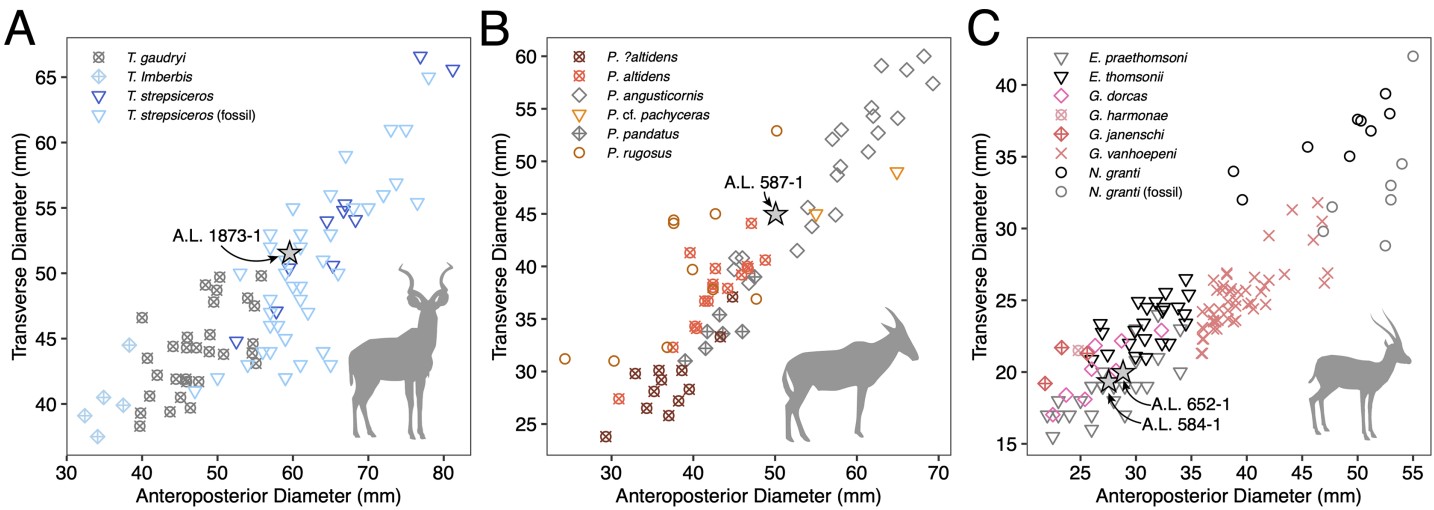

**Figure 4 Scatterplots of horn core anteroposterior and transverse dimensions.** (A) Kudus *Tragelaphus* spp. (B) *Parmularius* spp. (C) Male gazelles (*Eudorcas*, *Gazella*, and *Nanger* spp.). Data from *Bibi, Rowan & Reed (2017)*, *Gentry (1985, 2011)*, *Gentry & Gentry (1978)*, *Harris (1991a)*, and our own measurements.

**Description:** A.L. 1873-1 is a large left horn core preserving ~310 mm of its length from the base up (Figs. 3I and 3J). The horn core preserves some of the pedicel and is broken at the midfrontal suture, which allows orientation of the specimen. This break shows that the horns were inserted very close to the midline. Divergence is weak in anterior view. The pedicel is short, with the horn core surface dipping lower on the lateral side than on the medial side. The frontal shows no evidence of sinuses and is thick (midfrontal suture break = ~23.3 mm thick). The basal cross-section of the horn core is roughly oval and is transversely compressed (59.5 × 51.5 mm); there is some flattening of the posterolateral surface of the basal-most horn core. There is a weak anterior keel that is barely perceptible basally and that grows stronger distally; no other keels are present. Torsion is heteronymous. The horn completes a 180° twist and would have probably reached or surpassed a complete whorl (360°) when complete. The cross-section at the distal break is still large (43.3 × 32 mm). This horn is very similar in size, compression, and morphology to modern and fossil samples of *T. strepsiceros*, such as those from the Turkana Basin (Fig. 4A).

Three partial horn core midsections are also attributed to *T. strepsiceros*. A.L. 589-1 is a midsection (preserved length ~90 mm). The basal dimensions as preserved are 46.2 × 38.7 mm. A single keel, likely the anterior keel, is present. The cross-section is roughly oval and matches that of A.L. 1873. Torsion is present, with the horn completing a 20° twist in what is preserved. A.L. 586-2 is also a midsection (preserved length ~95 mm). The cross-section shows only one keel, presumably the anterior keel, and is oval with weak compression (basal dimensions as preserved are 42.5 × 37.9 mm). A.L. 596-4 is a longer midsection (preserved length ~165 mm); this also has an oval cross-section with compression and presence of a single keel; the horn completes a 90° twist. It is difficult to determine which end is proximal or distal, but the dimensions of cross-section are 43.5 × 38.8 mm at the middle of what is preserved.

A.L. 584-3 is a posterior cranium consisting mostly of the occipital bone. The size and morphology of this specimen (Figs. 3K–3N) match those of modern *T. strepsiceros* and Pleistocene specimens of *T. strepsiceros* from the Koobi Fora Formation (*Harris, 1991a*). This specimen lacks the supraoccipital torus that characterizes the bongo-like *T. nakuae*, which becomes especially prominent in specimens younger than ~2.3 Ma (*Bibi, 2011*). Because *T. nakuae* is not represented by horn cores from the Maka'amitalu, we attribute this cranium to *T. strepsiceros*.

Several partial mandibles, maxillae, and isolated teeth are recognized as tragelaphin based on their mesodonty, pointy cusps, simple central cavities, lack of goat folds, and lack of large basal pillars. Small basal pillars may be present on the lower molars. The dental sample is relatively homogenous in size (*e.g.*, m1 mesiodistal lengths 20.6 ± 1.62 mm, $n = 5$) and all specimens likely belong to *T. strepsiceros* (modern m1 mesiodistal lengths 20.8 ± 0.96, $n = 4$).

**Discussion:** The Maka'amitalu large tragelaphin remains can be differentiated from extinct *Tragelaphus moroitu* and extant *T. spekii* and *T. scriptus* by oval, transversely compressed cross-sections with only a single keel. They are larger, more widely divergent at the base, and have a more open spiral than those of the small kudu *T. gaudryi*, best known from similarly aged deposits in the Shungura Formation (*Gentry, 1985*; *Bibi, 2009*). Differentiating the Maka'amitalu remains from Pliocene *T. lockwoodi*, the likely ancestor of *T. strepsiceros*, is less straightforward, but the Maka'amitalu horns differ in lacking a triangular cross-section with keels in their distal parts (all of the preserved cross-sections, even the distalmost ones, are distinctly oval and possess a single keel). The retention of this feature in *T. lockwoodi* is primitive (*Reed & Bibi, 2011*) and the absence of these feature in the Maka'amitalu specimens indicates the evolution of a *T. strepsiceros*-like morphology. Furthermore, the width of the posterior cranium based on A.L. 584-3 (143 mm) is significantly larger than those of *T. lockwoodi*, for which *Reed & Bibi (2011)* gave a range of 92.4–107.4 mm, suggesting a shift to larger body size in these younger specimens.

The first appearance of *T. strepsiceros* is the Hata Member of the Bouri Formation in the Middle Awash ~2.5 Ma (*de Heinzelin et al., 1999*; *Bibi, 2009*). The Maka'amitalu specimens are therefore among some of the oldest greater kudu fossils known. Greater kudu are otherwise known from Olduvai Gorge (*Gentry & Gentry, 1978*) and the Turkana Basin (*Harris, Brown & Leakey, 1988*; *Harris, 1991a*; *Gentry, 1985*). This species is common in the fossil record from about ~2 Ma onwards (*Gentry, 2010*) and is widely distributed in the woodlands and thickets of eastern and southern Africa today.

*Tragelaphus* sp.

**Referred specimens:** A.L. 585-1, horn core.

**Description:** A.L. 585-1 is a basal horn core preserving some of the frontal. The horn core surface is eroded, but the cross-section is roughly triangular, with three keels (anterior, posteromedial, posterolateral) developed. Basal dimensions are 37.6e × 32e mm. Torsion is

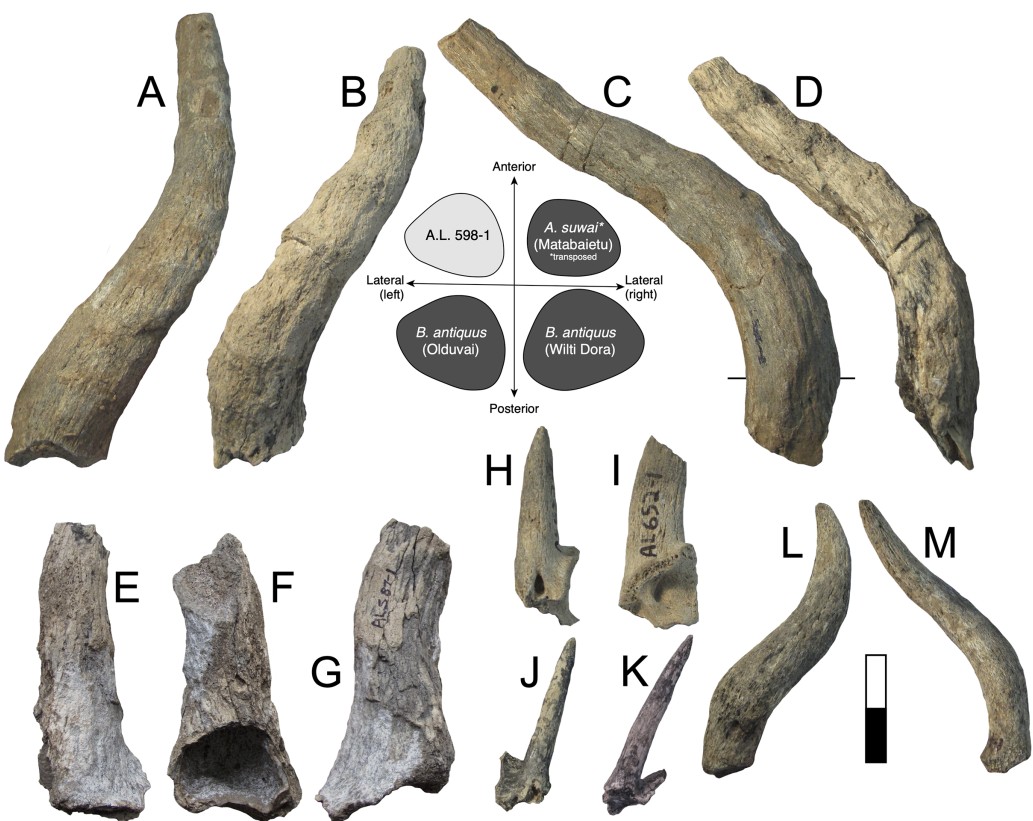

**Figure 5 Maka'amitalu Antilopinae.** A, C, A.L. 598-1 cf. *Beatragus antiquus* in anterior (A) and medial (C) view; B, D, A.L. 587-2 cf. *B. antiquus* in anterior (B) and medial (D) view. E–G, A.L. 587-1 *Parmularius altidens* or *P. angusticornis* in anterior (E), ?medial (F), and ?lateral (G) view. H-I, A.L. 652-1 *Eudorcas praethomsoni* in anterior (H) and medial (I) view; J–K, A.L. 666-3 *E. praethomsoni* in anterior (L) and lateral (M) view. L–M, A.L. 584-1 Antilopini gen. et. sp. indet. in ?lateral (N) and ?anterior (O) view. Scale bar equals 5 cm.

heteronymous. Weak cranial sutures and a relatively thin frontal suggest that the specimen was a juvenile.

**Discussion:** This poorly preserved horn core could belong to a small tragelaphin with primitive horn core morphology, such as the Shungura '*Tragelaphus pricei*' (*Gentry, 1985*). Even though it is likely a juvenile, it is too small and not sufficiently compressed to belong to *T. nakuae*.

Subfamily Antilopinae Gray, 1821
Tribe Alcelaphini Brooke in Wallace, 1876
cf. *Beatragus antiquus Leakey, 1965*

**Referred specimens:** A.L. 598-1, horn core; A.L. 587-2, horn core.

**Description:** A.L. 598-1 (Figs. 5A, 5C) is an alcelaphin horn core, probably from the left side, preserving most of its length (preserved length is ~310 mm measured along its course). None of the frontal is preserved, but the proximal end of the horn core shows the
superior part of a large smooth-walled sinus. Its basal dimensions are 41.1 × 51.8 mm, indicating weak compression along what is probably the anteroposterior axis. Assuming it is oriented correctly, there is marked posterior flattening, which causes the widest part of the horn core to occur posteriorly, giving the horn a roughly subtriangular basal cross-section. Horn core compression weakens distally so that the cross-section is almost circular at the end as preserved. Strong and well-spaced transverse ridges cover the anterior surface basally, although they wrap entirely around the horn core above the basal one-third. In anterior view, the horn is lyrate (similar in course to alcelaphins *Awashia*, *Beatragus*, or *Damaliscus lunatus*) with clear heteronymous torsion. In lateral view, the horn is straight above its insertion, but bends backward and straightens posteriorly above the basal one-third.

A.L. 587-2 is a horn core (Figs. 5B, 5D) clearly conspecific with A.L. 598-1. A.L. 587-2 is also likely from the left side, but otherwise is identical in size and morphology to the other specimen. This horn also shows weak basal anteroposterior compression (40e × 47.8 mm) and a preserved length of ~270 mm. Like A.L. 598-1, this horn is lyrate, is posteriorly flattened, and has well-spaced transverse ridges that are anterior basally but wrap around the horn core distally. The distal end has a near-circular cross-section. In lateral view the course is also similar, in that it possesses a straight basal portion that then courses posteriorly.

**Discussion:** These horn cores are similar in course to *Aepyceros*, but their basal sinuses are quite deep for an impala. Other differences from *Aepyceros* include anteroposterior compression, relatively triangular instead of rounded cross-sections, and absence of a posterolateral keel. Most species of *Damaliscus* have transversely compressed horn cores with flattened lateral surfaces, although the Olduvai species *Damaliscus agelaius* (Beds II–IV) has only weak basal horn core compression and possesses a flattened posterior surface rather than a lateral one. These features are shared with the Maka'amitalu specimens, but *D. agelaius* has shorter (mostly <250 mm) horn cores with smaller basal cross-sections that are virtually uncompressed (*e.g.*, 39.2 × 36.9 mm for the male skull VFK III–IV 214) (*Gentry & Gentry, 1978*).

*Vrba (1997)* erected *Awashia suwai* for a relatively complete alcelaphin skull from ~2.5 Ma sediments at Matabaietu in the Middle Awash. The Maka'amitalu horns share with the *A. suwai* holotype (MAT-VP-3/2) similar overall course and lyration, horn core length (reported as 280 and 300 mm for *Awashia*), well-spaced transverse ridges, and basal horn core dimensions with some flattening of the posterior horn core surface. Although *Vrba (1997)* reported only the holotype's minimum and maximum basal horn core dimensions (42.7 × 47.8 mm), our examination of this specimen confirms that these are very close to the anteroposterior and transverse measurements, respectively. The main differences between the Maka'amitalu horns and those of *Awashia* are the presence of stronger heteronymous torsion, stronger posterior curvature basally, and much more prominent transverse ridges.

Many of these features, especially well-developed heteronymous torsion, align the Maka'amitalu horns with Early Pleistocene *Beatragus antiquus* and extant *B. hunteri*.

 

*Vrba's (1997)* cross-sections of *B. antiquus* from Olduvai Gorge and Wilti Dora in the Middle Awash indicate a distinctly subtriangular shape with posterolateral flattening; the Maka'amitalu horns are a good match if rotated just slightly so that what is presumed to be the posterior surface becomes posterolateral (see cross-sections in Fig. 5). Basal horn core size and compression are variable in *B. antiquus* but the range of variation accommodates A.L. 598-1 and A.L. 587-2 especially given ambiguity in axis orientation (for example, at Olduvai, *Gentry & Gentry (1978)* provide the following for *B. antiquus*: 64.5 × 64.2, 50.9 × 41.5, 43.6 × 44.3, 65.8 × 67.2 mm). However, one of the clearest diagnostic characters of *B. antiquus* horn cores is a very long and straight distal portion (see plate 34 in *Gentry & Gentry, 1978*) and this cannot be assessed on the Maka'amitalu specimens. For this reason, we only tentatively attribute them to this taxon until more complete material is recovered. Records of *B. antiquus* are known from the KBS and Okote members of Koobi Fora (*Harris, 1991a*), Member G of the Shungura Formation (*Gentry, 1985*), Wilti Dora and Gamedah in the Middle Awash (*Vrba, 1997*), and Ahl al Oughlam in Morocco (*Geraads & Amani, 1998*).

Genus *Parmularius* Hopwood, 1934
*Parmularius altidens* Hopwood, 1934 or *P. angusticornis* Schwarz, 1937

**Referred specimens:** A.L. 587-1, horn core.

**Description:** A.L. 587-1 is a basal horn core of an alcelaphin the size of *Alcelaphus buselaphus* that cannot be confidently oriented (Figs. 5E–5G) although it is likely from the left side. Most of the horn core surface is eroded or poorly preserved and a break in its proximal portion, presumably on the medial side, shows that the basal horn core possessed a large smooth-walled sinus penetrating deep into the horn core proper. The horn core pedicel is tall and straight as in *Parmularius* spp. (in contrast to *Damaliscus* spp.). Its basal portion is straight, above which the horn bends slightly laterally and more strongly posteriorly. The basal cross-section is not very compressed (basal dimensions are 50e × 45e mm), with an irregularly rounded cross-section.

**Discussion:** This poorly preserved specimen is reminiscent of various *Parmularius* species from Olduvai Gorge (*Gentry & Gentry, 1978*), especially the *Parmularius altidens-P. angusticornis* lineage. A.L. 587-1 can be referred to *Parmularius* based on its overall size (roughly the size of a hartebeest), long horn core pedicel, weak transverse compression and irregular basal cross-section (though any 'swellings' have been eroded away), and lack of keels or torsion (*Gentry & Gentry, 1978*; *Gentry, 2010*). It shares with *P. altidens* and *P. angusticornis* a fairly straight basal horn core course with gentle backward curvature and perhaps a more or less localized lateral divergence with *P. angusticornis*. Its basal horn core dimensions fall towards the highest end of the Olduvai range of variation for *P. altidens*. Mean dimensions of the middle Bed I assemblage from FLKN I are 42.3 × 42.3 mm, but the largest specimen has an anteroposterior diameter of 50.3 mm and transverse diameters of 44.5 mm, which fits the Maka'amitalu horn core. A more comfortable fit metrically is with the younger (Bed II) species *P. angusticornis*, which has

average basal horn core dimensions of 57.7 × 49.6 mm (anteroposterior diameter range is 45–69.3 mm and transverse diameter range is 38.4–60 mm in *Gentry & Gentry, 1978*). Metric comparisons with other eastern African *Parmularius* species are shown in Fig. 4B. However, *P. altidens* and *P. angusticornis* differ in other features, namely the length and course of the horns and the degree of basal swelling, features that unfortunately cannot be assessed on A.L. 587-1.

*Parmularius altidens* might otherwise be known from the Early Pleistocene KBS Member of Koobi Fora and Members G-H of the Shungura Formation (*Gentry, 1985*; *Harris, 1991a*) and the latest Pliocene of the Upper Ndolanya Beds of Laetoli (*Gentry, 2011*). *Parmularius angusticornis* is restricted to the Early Pleistocene of Olduvai Bed II and nearby localities, such as Peninj and Kanjera (*Gentry, 2010*). The Bed I-Bed II Olduvai record suggests these two species are parts of an anagenetic lineage (*Gentry & Gentry, 1978*).

Alcelaphini gen. et. sp. indet.

**Referred specimens:** A.L. 586-1, right P3; A.L. 586-9, right mandible with p4-m1; A.L. 588-5, left P4; A.L. 590-2, left mandible with m1-m2; A.L. 591-2, right m1/m2; A.L. 592-6, left mandible with dp4; A.L. 594-3, left M3; A.L. 594-4, right M3; A.L. 595-3, right M3; A.L. 596-2, right dP4; A.L. 597-1, right M1/M2; A.L. 658-2, right M1/M2; A.L. 704-1, left m1/m2; A.L. 753-1, left m1/m2.

**Description:** These are alcelaphin teeth that cannot be identified below the tribal level. They are recognizable as alcelaphin based on their hypsodonty, possession of cement, rounded lingual lobes of the upper molars and buccal lobes of the lower molars, and lack of basal pillars and goat folds (*Gentry & Gentry, 1978*).

**Discussion:** These dental remains probably represent more than one species based on size variation (*e.g.*, m1 mesiodistal lengths range 17.9–23.5 mm), and many may belong to species described above based on horn core remains.

Tribe Antilopini Gray, 1821
Genus *Eudorcas* Fitzinger, 1869
*Eudorcas praethomsoni* (Arambourg, 1947)

**Referred specimens:** A.L. 652-1, horn core; A.L. 666-3, horn core; A.L. 666-6, horn core; A.L. 754-1, horn core. A.L. 1877-1, right mandible with m3; A.L. 653-5, right maxilla with P4-M2; A.L. 659-1, left mandible with p2-m3; A.L. 659-3, left mandible with p3-m2; A.L. 666-14, right p3; A.L. 666-17, left maxilla with P4-M2; A.L. 666-18, right maxilla with P2-P3; A.L. 666-22, left mandible with m2; A.L. 666-24, right maxilla with M2-M3 and isolated M3.

**Description:** A small gazelle is represented by several horn cores. A.L. 652-1 is a small, presumably male, basal horn core preserving some of the frontal (Figs. 5H and 5I). It is strongly compressed transversely (basal dimensions are 27.5 × 19.4 mm) and the lateral surface is flattened. A large and triangular supraorbital foramen occurs just below the horn

core pedicel; the horn core to pedicel transition is smooth and without lipping. The horn core insertions were very close to the midline.

A.L. 754-1 is a basal section of horn core preserving a very small piece of pedicel. The horn core surface-pedicel transition is better marked than in A.L. 652-1, but otherwise these horns are very similar in morphology. Basal dimensions are 28.8 × 20e mm. A.L. 666-6 is a mid-section of horn core that is larger than A.L. 652-1 and 754-1 but is otherwise similar in morphology and compression. Dimensions at its basal break are 34.3 × 24.2 mm and its preserved length is ~110 mm. While this could indicate the presence of a larger-bodied gazelle species in the assemblage, the antilopin dental sample shows little size variation, and we conservatively include A.L. 666-6 here.

A.L. 666-3 is a partial frontlet with a complete horn core (length = 65 mm) that is very slender and round and likely represents a female of this species (Figs. 5J and 5K). The horn core is straight in lateral view, with only a hint of backward curvature. There is flattening of the lateral surface and some of the posterior surface, with a very weak posterolateral keel developed basally between these two surfaces. Its basal cross-section (12.8 × 12.1 mm) is similar in size to modern female *Gazella dorcas*. It was closely inserted to the midline, as is the case in the male specimens, and the horn core surface-pedicel transition is also weak. There is a moderate postcornual fossa. The supraorbital pit is not easily identifiable because of some erosion of the frontal.

The Maka'amitalu horn cores are too compressed to belong to '*Gazella*' *janenschi* despite being of similar size (we place the genus in quotes as the relationships between these different fossil species and extant *Gazella* spp. are undemonstrated and often doubtful). They are much smaller than those allied with *Nanger granti* from Koobi Fora (*Harris, 1991a*) and Laetoli (*Gentry, 2011*). The same is true of the large South African species '*Gazella*' *vanhoepeni* from Makapansgat (*Wells & Cooke, 1956*), including its females '*G. gracilior*' (*Locke, Rowan & Reed, 2016*); additionally, the Makapansgat species has quite extensive sinuses that penetrate deep into the horn core which sheds doubt on any relationship to extant gazelles. The Maka'amitalu specimens are very different from '*G.*' *harmonae*, known from the Hadar Formation, which possesses a weakly compressed and roughly round basal cross-section of its horns (*Geraads, Bobe & Reed, 2012*). The best match for the Maka'amitalu specimens is '*Gazella*' *praethomsoni* from the Koobi Fora (*Harris, 1991a*) and Shungura formations (*Gentry, 1985*) (Fig. 4C). They share with this species small and strongly compressed horn cores with flattening of the lateral horn core surface.

The size of antilopin dentitions from the Maka'amitalu is homogenous and all probably belong to this species. These teeth are identified as antilopin based on their small size, tall crowns, lack of basal pillars, pointy cusps, and lack of paraconid-metaconid fusion on p4, differentiating them from *Aepyceros*. They are slightly smaller than the teeth *Harris (1991a)* referred to '*G.*' *praethomsoni* from Koobi Fora but are similar to modern *Eudorcas thomsonii* (*e.g.*, m3 mesiodistal lengths are 17.5 and 18.5 mm for the Maka'amitalu, 18 ± 0.64 for *E. thomsonii* ($n$ = 6), and 20.6 and 20.9 mm for two specimens from Koobi Fora).

**Discussion:** The African fossil record of gazelles is scrappy and virtually all remains have been attributed to the genus *Gazella*, which historically includes all extant species. There is now good evidence from phylogenies based on morphological (*Groves, 2000*), chromosomal (*Cernohorska et al., 2015*), and genomic data (*Bärmann, Rössner & Wörheide, 2013*) that this grouping is paraphyletic. Recognizing this, *Groves (2000)* resurrected the names *Eudorcas* Fitzinger 1869 and *Nanger* Lataste 1885 for the Thomson's and Grant's gazelle groups, respectively. The genus *Eudorcas* contains *Eudorcas thomsonii* (plus *E. albonotata*) and *E. rufifrons* (plus *E. tilonura*), whereas *Nanger* contains *Nanger granti*, *N. dama*, and *N. soemmerringi* (*Groves & Grubb, 2011*).

Though no systematic revision of fossil Antilopini has recently been undertaken, '*Gazella*' *praethomsoni*, known primarily from the Turkana Basin, has been consistently aligned with *Eudorcas thomsonii* and *E. rufifrons* (*Gentry, 2010*; *Geraads et al., 2004*; *Harris, 1991a*). Specimens of '*G.*' *praethomsoni*, including those from Maka'amitalu, share with extant *Eudorcas* species short and transversely compressed horn cores with little lateral divergence and backward curvature. *Groves & Grubb (2011)* claim that stronger transverse compression differentiates horns of *Eudorcas* from those of similarly sized *Gazella* species. Our data for male *E. thomsonii* (compression index $0.75 \pm 0.05$, $n = 22$) and male *G. gazella* ($0.82 \pm 0.03$, $n = 6$) support this distinction, although the differences are less marked when compared to *G. dorcas* ($0.76 \pm 0.04$, $n = 8$) or *G. spekei* ($0.78 \pm 0.01$, $n = 2$). However, transverse compression in '*G.*' *praethomsoni* is even stronger ($0.69 \pm 0.06$, $n = 27$) than in living *Eudorcas* species. Likewise, both *Eudorcas* and '*G.*' *praethomsoni* are significantly smaller than extant *Nanger* species and the larger gazelles allied with *N. granti* in the eastern African fossil record (Fig. 4C). It therefore is reasonable to attribute '*G.*' *praethomsoni* to *Eudorcas*, as it is clearly near the ancestry of, if not directly ancestral to, living *Eudorcas* species. The differences that separate *E. praethomsoni* from *E. thomsonii* essentially amount to the former's smaller horn cores with stronger transverse compression and posterior curvature. These differences could be the result of size allometry, such that an increase in body size alone might account for the associated horn core differences.

Antilopini gen. et. sp. indet.

**Referred specimens:** A.L. 584-1, a horn core preserving a small amount of frontal.

**Description:** A.L. 584-1 is a complete horn core (~130 mm long) but cannot be oriented because not enough of the frontal is preserved (Figs. 5L and 5M). The frontal is very thick (~15.5 mm) for a bovid of this size and the pedicel is short. The basal horn core is quite compressed (26.8 × 20.6 mm) though it is uncertain in which direction this is. In ?lateral view, the horn core bends backward immediately over its insertion and then recurves distally giving it a sigmoid shape with some torsion evident towards its tip. In ?anterior view, the horn diverges laterally but then straightens in its distal third; faint transverse ridges occur basally, but vanish distally. There is some longitudinal grooving of the ?posterior surface.

**Discussion:** Identification of this specimen is confounded by its lack of a base. It is very likely to be antilopin based on its small size combined with lack of sinuses, keels, or deep longitudinal grooving. Such horn core curvature is reminiscent of *Antidorcas* fossil specimens, but the lack of sinuses precludes this genus.

Tribe Hippotragini Sundevall, 1845
cf. *Oryx* sp.

**Referred specimens:** A.L. 589-3, left mandible with p3-m3; A.L. 591-10, right M3; A.L. 595-1, right mandible with p3-m3; A.L. 599-1, partial dentition (associated upper and lower teeth); A.L. 652-4, left mandible with p3-p4; A.L. 653-8, left mandible with dp2-m1; A.L. 657-1, left mandible with p3-p4; A.L. 658-3, left m1/m2; A.L. 1874-1, right m3; A.L. 1880-2, left mandible with m2-m3.

**Description:** Dental remains were identified as hippotragin based on the expanded and bulbous metaconid on p4, and molars with strong goat folds (lowers) and lack of pinched lobes (buccal on lowers, lingual on uppers). The most informative specimens are two mandibles, A.L. 589-3 and A.L. 595-1, both of which preserve p3-m3. Although overlapping in molar size with the large-bodied *Kobus*, these specimens are definitely hippotragin based on p4 morphology; both mandibles have p4s with very large, rounded metaconids and lack the projecting hypoconid typical of reduncins.

**Discussion:** The hippotragin dental sample is homogenous in size and is comparable to modern *Oryx beisa* (*e.g.*, m3s for Maka'amitalu are 30.1 ± 0.63 mm (*n* = 5) and 29.84 ± 1.24 mm for *O. beisa* (*n* = 9)). They are similar in size to a hippotragin m3 from the KBS Member of the Koobi Fora Formation (mesiodistal length = 31.8 mm) that *Harris (1991a)* referred to *Oryx*. Seeing as these teeth are smaller than those of *Hippotragus gigas* (*Gentry & Gentry, 1978*), the only other Early Pleistocene hippotragin known from eastern Africa, we tentatively refer them to *Oryx*.

Tribe Reduncini Knottnerus-Meyer, 1907
Genus *Kobus* A. Smith, 1840
*Kobus sigmoidalis* Arambourg, 1941 or *K. ellipsiprymnus* (Ogilby, 1833)

**Referred specimens:** A.L. 586-12, right mandible with dp4-m1; A.L. 586-13, left mandible with p2-m3; A.L. 587-4, left mandible with p4-m3; A.L. 666-23, right maxilla with M1-M2; A.L. 591-6, right m3 fragment.

**Description:** These are waterbuck-sized reduncin dental remains that fit comfortably within the range of modern *Kobus ellipsiprymnus* and its probable ancestor *K. sigmoidalis*. For example, the m1-m3 lengths of the Maka'amitalu specimens (65.6 and 66 mm) are similar to those of *K. sigmoidalis* from Olduvai Bed I (68.6 ± 5.4 mm, *n* = 5) and Koobi Fora (61.2 ± 6.4 mm, *n* = 3) and the living waterbuck (69.3 ± 0.92 mm, *n* = 6) as well as the Late Pleistocene fossil remains of this species from the Kibish Formation (*Gentry & Gentry, 1978*; *Harris, 1991a*; *Rowan et al., 2015*).

**Discussion:** In the Early Pleistocene, *Kobus sigmoidalis* or *K. ellipsiprymnus* would be the most likely attribution for such large reduncin teeth. This lineage was widespread in eastern Africa and has been found at Olduvai Gorge (*Gentry & Gentry, 1978*), the Turkana Basin (*Gentry, 1985*; *Harris, Brown & Leakey, 1988*; *Harris, 1991a*), and the Afar (*de Heinzelin et al., 1999*; *Bibi, Rowan & Reed, 2017*). It is best known from the Shungura Formation, especially Members C-G. *Gentry (1985)* noted a size increase through time in the Shungura sequence with *K. sigmoidalis* appearing to give rise to the living waterbuck in Member G, though *Vrba (2006)* believed *K. oricornus* to be a more suitable ancestor for *K. ellipsiprymnus*.

cf. *Redunca* or *Kobus kob*

**Referred specimens:** A.L. 584-2, left mandible with p4-m1; A.L. 591-7, mandible with right and left dp4-m2; A.L. 591-8, right mandible with m1-m3.

**Description:** These are reduncin dental remains too small to belong to the larger waterbuck-sized species referred to *Kobus sigmoidalis* or *K. ellipsiprymnus*.

**Discussion:** This smaller species is metrically a better match for *Redunca* than *Kobus kob* (m1 mesiodistal lengths are $10.9 \pm 1.1$ mm for Maka'amitalu ($n = 4$), $11.8 \pm 0.5$ mm for *Redunca arundinum* ($n = 3$), $10.5 \pm 0.63$ mm for *R. fulvorufula* ($n = 5$), $10.9 \pm 1.1$ mm for *R. redunca* ($n = 5$), and $12.5 \pm 0.8$ mm for *K. kob* ($n = 5$)) but there is much overlap among these taxa.

Tribe Indeterminate
Bovidae gen. et. sp. indet.

**Referred specimens:** A.L. 586-3b, right mandible with ?dp4; A.L. 590-3, a crushed occipital; A.L. 1882-2, incisor.

**Description:** These remains include a small occipital that is dorsoventrally crushed. Two dental specimens are also unidentifiable below the family level.

**Discussion:** The size of the occipital would fit *Eudorcas praethomsoni* but is badly crushed and uninformative as to tribe. The only other bovid remains from locality A.L. 590 are alcelaphin dentitions and a horn core of *Syncerus acoelotus*, both of which are much too large to fit the occipital.

Family Giraffidae Gray, 1821
Subfamily Giraffinae Gray, 1821
Genus *Giraffa* Brisson, 1762
*Giraffa* sp.

**Referred specimens:** A.L. 591-9, left p4; A.L. 652-2, left p4 fragment.

**Description:** A species of *Giraffa* is represented by two p4s, both from the left side. Their smaller size and brachydonty differentiate them from premolars of *Sivatherium*, and they are too molarized to represent *Palaeotragus*.

**Discussion:** *Giraffa* is rare in the Maka'amitalu fauna. The size of A.L. 591-9 is similar to p4s of *Giraffa jumae*, but more complete remains would be needed to confirm this identification.

Subfamily Sivatheriinae Murie, 1871
Genus *Sivatherium* Falconer and Cautley, 1836
*Sivatherium maurusium* (Pomel, 1892)

**Referred specimens:** A.L. 584-5, upper molar fragment; A.L. 666-25, upper molar fragment.

**Description:** Two giraffid upper molar fragments can be identified as *Sivatherium* based on their large size and hypsodonty.

**Discussion:** As with *Giraffa*, *Sivatherium* is rare in the Maka'amitalu fauna. These remains are attributed to *Sivatherium maurusium*, the common Plio-Pleistocene sivathere in eastern and southern Africa. *Sivatherium hendeyi* is an earlier species known from the earliest Pliocene of Langebaanweg, South Africa. Some authors have contested the separation of *S. hendeyi* from *S. maurusium* (*e.g.*, Churcher, 1978).

Family Hippopotamidae Gray, 1821
Subfamily Hippopotaminae Gray, 1821
Genus *Hippopotamus* Linnaeus, 1758
*Hippopotamus* cf. *gorgops* Dietrich, 1928

**Referred specimens:** A.L. 593-1, right calcaneum; A.L. 593-2, right astragalus.

**Description:** A.L. 593-1 and A.L. 593-2 (Figs. 6A–6E) are somewhat damaged: the fibular process of the calcaneum is missing and the tuberosity is eroded; the astragalus plantar surface is also eroded distally. Yet, most measurements could be acquired. Although their anatomical features do not depart from the general pattern observed in Hippopotamidae, they are both remarkable in displaying large size and great robustness, and the two specimens could belong to a single individual. The astragalus is relatively and absolutely wider than any specimen from the Hadar Formation at Hadar. The calcaneum is also significantly longer and more robust than those from the Hadar Formation. In fact, compared with data gathered by one of us (J.-R.B.) for Hippopotamidae during the Neogene and Quaternary, the Maka'amitalu calcaneum is the largest of all fossil and extant specimens ($n = 113$), being approached in size only by a specimen from Member K in the Shungura Formation (Fig. 6F). Similarly, in comparisons with over 660 specimens (Fig. 6G), the Maka'amitalu astragalus matches only the dimensions observed in less than a dozen very large specimens from Konso, from the upper sequence of the Omo Group in

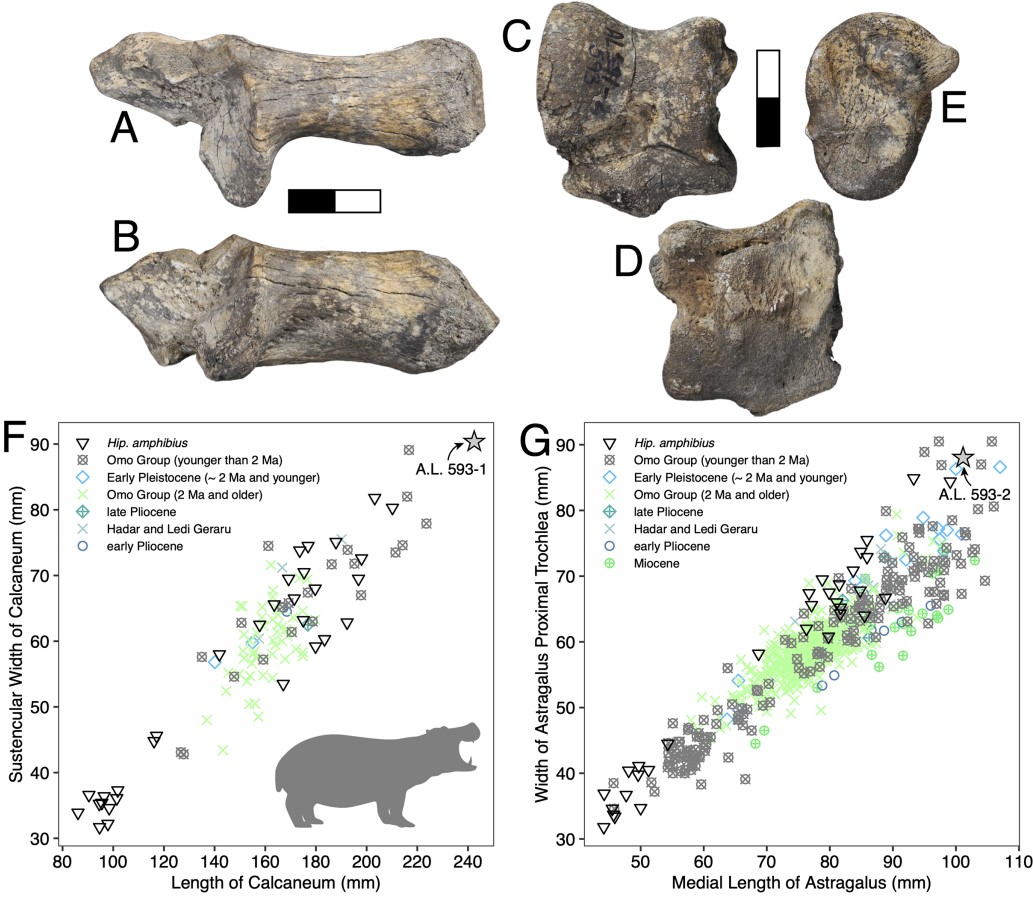

**Figure 6 Maka'amitalu *Hippopotamus* cf. *gorgops*.** A.L. 593-1 in proximal (A) and medial (B) views. C–E, A.L. 593-2 in anterior (C), palmar (D), and medial (E) views. (F) Scatterplot of calcaneum length *vs* width at sustentaculum in Hippopotaminae (our personal measurements). (G) Scatterplot of astragalus medial length *vs* proximal trochlear width in Hippopotaminae (including data from *Harris, 1991b* and our personal measurements); Scale bar equals 5 cm.

the Turkana Basin, from the Middle Pleistocene of the Middle Awash, and extant *Hippopotamus amphibius*.

**Discussion:** Specific attribution of isolated hippopotamid astragali and calcanei is usually challenging because of the apparent lack of interspecific variation in these bones, at least in terms of discrete features. A thorough study of hippopotamid postcranial elements still needs to be performed. However, extreme dimensions and proportions can be informative, and the Maka'amitalu hippopotamid specimens definitely fall in this category. The proportions of the astragalus A.L. 593-2 are not known in hippopotamine primitive lineages, such as aff. *Hippopotamus* from the Turkana Basin or the species described in the Afar Depression prior to the BKT-2 deposition (see *Gèze, 1980*). In the Plio-Pleistocene of Africa, these proportions and dimensions are found only in specimens attributed to *H. gorgops* Dietrich, 1928, including specimens found in association with dental material (*e.g.*, KNM-ER 2279 at Koobi Fora, Kenya; see *Harris, 1991b*). The dimensions of the Maka'amitalu tarsals are overall congruent with the particularly large dimensions of the

craniodental material attributed *H. gorgops*. It can be noted that such dimensions and proportions are also known in other species of *Hippopotamus*, notably in very large specimens of the extant *H. amphibius* Linnaeus, 1758. In absence of further hippopotamid remains at Maka'amitalu, we attribute the Maka'amitalu tarsals to *H.* cf. *gorgops*.

Very large representatives of *Hippopotamus*, most often identified as *H. gorgops*, are well known in eastern Africa at Olduvai Gorge (*Coryndon, 1970*), but also in the Turkana Basin (*Harris, Brown & Leakey, 1988*; *Harris, 1991b*), in the Middle Awash (*Boisserie & Gilbert, 2008*), and at Konso. In the Turkana Basin, they appear at ~2 Ma, at the transition between the lower and upper parts of Member G in the Shungura Formation and in the upper Burgi Member of the Koobi Fora Formation. In other sites, they occur in more recent sediments. This suggest either that the Maka'amitalu postcranial remains record the earliest occurrence of very large *Hippopotamus* in eastern Africa, or that the Maka'amitalu hippopotamid remains are 2 Ma or younger. Given the extensive records of fossil hippopotamids from the Afar and Turkana Basin, the latter hypothesis is most parsimonious.

Family Suidae Gray, 1821
Subfamily Suinae Gray, 1821
Genus *Kolpochoerus* Van Hoepen and Van Hoepen, 1932
*Kolpochoerus* cf. *phillipi* *Souron, Boisserie & White, 2015*

**Referred specimens:** A.L. 1908-1, a partial cranium with right P4-M3, left C-M3, and other fragments; A.L. 588-1A, left M3; A.L. 588-1B, right M3; A.L. 588-2, left i2; A.L. 588-4, left M2 fragment; A.L. 589-2, left I1; A.L. 591-5, left c fragment.

**Description:** A.L. 1908-1 is the most complete suid specimen recovered from Maka'amitalu (Figs. 7A and 7B). This partial cranium preserves P4-M3 on the right side and P2-M3 on the left side, as well as the base of the left canine. Cranial elements preserved in association with the maxilla include the ventral part of the palate, the left supracanine flange, other fragments of the maxilla, premaxilla, and a small portion of the zygomatic or frontal bone near the orbital margin. The teeth were covered by a layer of hard carbonate and the removal of this matrix revealed poorly preserved occlusal surfaces. Though the general morphology of the dentition is reminiscent of *Kolpochoerus afarensis* or early specimens of *K. limnetes*, the mesiodistal compression of the lateral pillars on the upper molars clearly align this specimen with the *K. phillipi-K. majus* lineage. The dental morphology of A.L. 1908-1, particularly with respect to the M3, is comparable to the holotype of *K. phillipi* from Matabaietu (*Souron, Boisserie & White, 2015*) and the earliest *K.* cf. *majus* from Konso (*Suwa, Souron & Asfaw, 2014*). Mesiodistal compression is particularly evident on the labial pillars of the A.L. 1908-1 M3s. The M3 talons are relatively simple, consisting of two to three moderately large pillars.

The size of the M3s of A.L. 1908-1 ($38.9 \times 23.4$ mm, $38.5 \times 23.7$ mm) fall within the range of *K. afarensis* ($35.7 \pm 3.0 \times 22.0 \pm 1.4$ mm, $n = 19$) but are closer to the average of *K. majus* ($39.5 \pm 4.0 \times 23.1 \pm 1.8$ mm, $n = 50$), and only slightly larger than MAT-VP-1/5 ($36.8 \times 22.4$ mm) (Fig. 8). The P3 and P4s of A.L. 1908-1 are similar to those of other

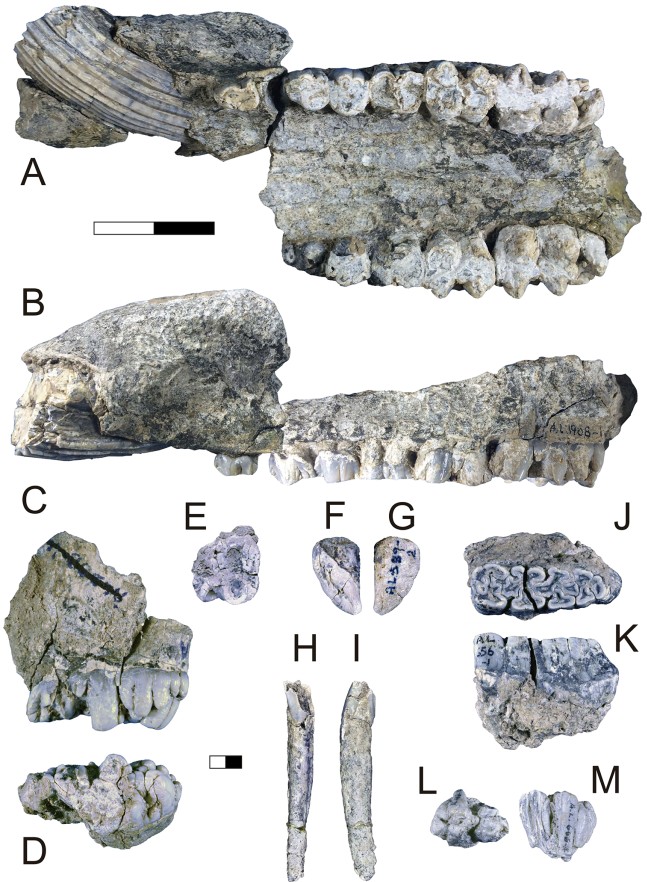

**Figure 7 Maka'amitalu Suidae.** (A and B) A.L. 1908-1 *Kolpochoerus* cf. *phillipi* maxilla in occlusal (A) and lateral (B) views. (C and D) A.L. 588-1A *K.* cf. *phillipi* left M3 in buccal (C) and occlusal (D) views. (E) A.L. 588-4 *K.* cf. *phillipi* mesial fragment of a left M2 in occlusal (E) view. (F and G) A.L. 589-2 *K.* cf. *phillipi* left I1 in lingual (F) and labial (G) views. (H and I) A.L. 588-2 *K.* cf. *phillipi* right i2 in lingual (H) and mesial (I) views. (J and K) A.L. 656-1 *Metridiochoerus modestus* right m3 in occlusal (J) and lingual (K) views. (L and M), A.L. 666-16 *M. modestus* right M2 in occlusal (L) and buccal (M) views. Scale bar equals for (A and B) equals 40 cm and scale bar for (C–M) equals 10 cm.

kolpochoeres and their size (P3, 14.7 × 14.0 mm; P4, 13.5 × 16.9 mm) falls within the range of *K. afarensis* (P3, 14.6 ± 1.6 × 12.1 ± 3.3 mm, *n* = 2; P4, 13.7 ± 0.8 × 15.7 ± 1.3 mm, *n* = 6). They are smaller than *K. majus* (P3, 16.4 ± 2.0 × 16.1 ± 2.1 mm, *n* = 15; P4, 15.4 ± 1.8 × 17.7 ± 2.1 mm, *n* = 25), and slightly larger than MAT-VP-1/5 (P3, 9.2 × 13.3 mm; P4, 9.6 × 14.0 mm). The supracanine flange is exceptionally well-developed and, together with the large size of the canine, suggests that the Maka'amitalu palate belonged to a male. Development of the flange is similar to that of the *K. phillipi* holotype (*Souron, Boisserie & White, 2015*) and one *K.* cf. *majus* cranium from Konso (*Suwa, Souron & Asfaw, 2014*). The supracanine flange is also well-developed in younger *K. majus* material from Daka (~1 Ma) but the crest is not as well defined and not separated by a deep furrow as in the holotype of *K. phillipi* and the *K.* cf. *majus* material from Konso (*Gilbert, 2008b*; *Suwa, Souron & Asfaw, 2014*).

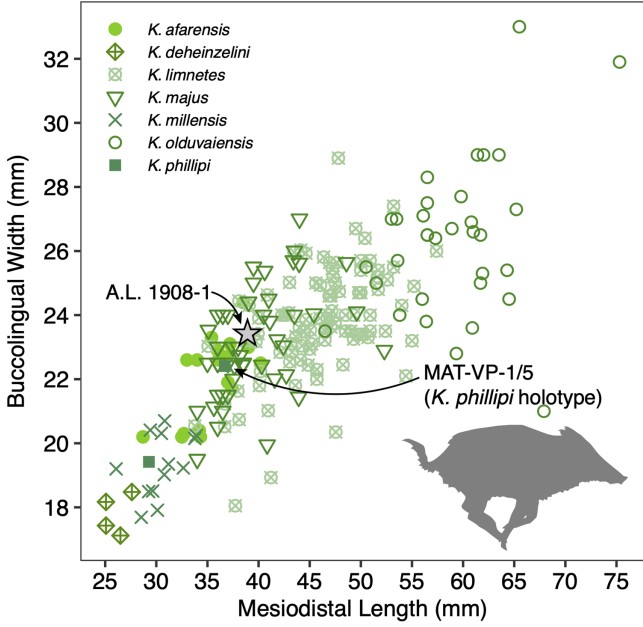

**Figure 8 Scatterplot of *Kolpochoerus* M3 mesiodistal lengths and buccolingual widths.** Data from *Brunet & White (2001)*, *Gilbert (2008b)*, *Souron, Boisserie & White (2015)*, *Suwa, Souron & Asfaw (2014)*, and our personal measurements.

Other specimens referred to *K.* cf. *phillipi* from the Maka'amitalu include A.L. 588-1A, A.L. 588-1B, and A.L. 588-4 (Figs. 7C–7E), which could belong to the same individual based on morphology and wear stage. A.L. 588-1A is an almost complete left M3 with moderate wear and is broken mesially (missing the entire protocone). In addition to the four main mesial pillars, A.L. 588-1A displays a set of three independent and well-defined accessory cusps in the distalmost labial part of the tooth supplementing two basal labial cusps and forming a rim of cusps that surrounds the distal part of the tooth (see Figs. 7C and 7D). This feature is variably present in derived specimens of *K. majus*, such as those from Olorgesailie (see *Harris & White, 1979*: fig. 68), but has yet to be recorded in the hypodigm of *K. phillipi* or in the *K. limnetes-K. olduvaiensis* lineage.

In this regard, A.L. 588-1A is more derived than A.L. 1908-1 in having greater complexity of labial accessory cusps. Its basal length is estimated at 41.8 mm, which is larger than the M3 from the *K. phillipi* holotype MAT-VP-1/5, larger than all of the *K.* cf. *majus* from the Interval 1 of Konso (~1.9 Ma) (*Suwa, Souron & Asfaw, 2014*), but within the range specimens from Interval 4 and Interval 5 (~1.4 Ma) and younger *K. majus* (*Suwa, Souron & Asfaw, 2014*; *Souron, Boisserie & White, 2015*). A.L. 588-1B is the right M3 of the same individual but is less complete, as it is missing all of its mesial portion (protocone and paracone). A.L. 588-4 is a mesial fragment of a left M2 that is compatible in morphology (mesiodistal compression of pillars) and wear with A.L. 588-1A and A.L. 588-1B.

We tentatively refer three additional specimens to *K.* cf. *phillipi*. A.L. 589-2 is a lightly worn left I1 that preserves most of its crown but is missing the entire root (Figs. 7F and 7G). The morphology and size (21.5 × 26.6 mm) of this tooth are compatible with *K. phillipi* or *K. majus*, but only one first upper incisor is known from *K. phillipi*

(*Souron, Boisserie & White, 2015*: fig. 3A4) and this tooth is too worn for appropriate comparison. A.L. 588-2 is a right i2 that preserves a complete root but is missing most of the crown (Figs. 7H and 7I). The tooth is narrow (9.7e × 11.3 mm) as in *Kolpochoerus*. The labial enamel surface is longer (~10 mm) than the lingual one at the base, a character that is typical of modern *Sus* or *Potamochoeru*s (*Lazagabaster, 2013*). Finally, there is a small fragment of the tip of a lower canine, A.L. 591-5.

**Discussion:** Originally attributed to *Kolpochoerus* cf. *limnetes* (*Kimbel et al., 1996*), the most complete remains of *Kolpochoerus* from Maka'amitalu likely represent *Kolpochoerus phillipi*, a species erected by *Souron, Boisserie & White (2015)* based on a relatively complete skull from ~2.5 Ma deposits at Matabaietu in the Middle Awash. Other remains attributed to *K. phillipi* have been recently described from the Gurumaha (~2.84–2.75 Ma) and Lee Adoyta (~2.7–2.6 Ma) fault blocks at Ledi-Geraru (*Lazagabaster et al., 2018*). Unfortunately, it seems that very complete craniomandibular material is required to confidently identify remains as *K. phillipi* (*Souron, Boisserie & White, 2015*), which is not the case for the Maka'amitalu sample. The only other diagnostic character for this species may be in the molars, which possess mesiodistally compressed lateral pillars compared to *K. afarensis* or *K. limnetes*, but not as extreme as those of derived *K. majus* specimens. From Mille-Logya, *Geraads et al. (2021)* listed Early Pleistocene kolpochoere material as *Kolpochoerus* sp., suggesting that isolated teeth of *K. phillipi* cannot be easily distinguished from those of *K. limnetes*. However, the mesiodistal compression of the Maka'amitalu dentitions is obvious and comparable to that the Matabaietu sample of *K. phillipi*, whereas *K. limnetes* tends to have longer and more complex third molars than *K. phillipi*. In our view, the Mille-Logya specimens attributed to *Kolpochoerus* by *Geraads et al. (2021)* likely represent early forms of *Metridiochoerus*. For example, the M3 with specimen number MLP-2655 (*Geraads et al., 2021*: Fig. 3Q) displays a relatively high crown, main cusps with relatively flat sides, and accessory cusps that are relatively tall and columnar.

The phylogenetic hypothesis of *Souron, Boisserie & White (2015)* places *Kolpochoerus phillipi* in the temporal and morphological gap between *K. afarensis*, a suid common in the Afar between 3.5–2.95 Ma, and *K. majus*, a typical eastern African Pleistocene suid with a tendency towards increased molar bunolophodonty. It is thought that *K. majus* eventually gave rise to the extant giant forest hog *Hylochoerus meinertzhageni* (*Souron, Boisserie & White, 2015*). Collectively, these taxa represent a 'bunolophodont' lineage separated from the *K. limnetes*-*K. olduvaiensis* lineage. Because *K. phillipi* and *K. majus* are likely chronospecies, delimiting a late specimen of the former from an early specimen of the latter is not straightforward. Based on the diagnosis of *Souron, Boisserie & White (2015)*, *K. phillipi* appears to be only clearly distinguished from derived *K. majus* younger than 1 Ma. This ambiguity led *Suwa, Souron & Asfaw (2014)* to attribute the material from Konso to *K.* cf. *majus* as it is both older and more primitive than younger *K. majus* specimens. The Maka'amitalu specimens likewise may represent a transitional form with intermediate morphology between the type material of *K. phillipi* and the specimens from Konso. For example, the upper third molars have more elaborated talons and are larger than most specimens attributed to *K. phillipi* and fall within the range of *K. majus* in both

respects. Thus, the similarity to older and younger specimens makes attribution to one or the other chronospecies difficult.

Genus *Metridiochoerus* Hopwood, 1926
*Metridiochoerus modestus* (Van Hoepen and Van Hoepen, 1932)

**Referred specimens:** A.L. 656-1, right m3; A.L. 666-16, right M2; A.L. 666-19, fragment of left m2; A.L. 703-1, left mandible with roots of p4, fragment of m1, and m2.

**Description:** A.L. 656-1 is a complete right m3. It is well preserved except for a crack (~0.6 mm wide) that runs labiolingually across its middle portion (Figs. 7J and 7K). The tooth is moderately worn, WS 7 in the classification of *Kullmer (1999)*, and clearly would have been quite hypsodont. The morphology of the occlusal surface is intermediate between *Metridiochoerus andrewsi* and later forms of *M. modestus*. There are four main pairs of lateral pillars, terminated by a well-developed pillar accompanied by a very small columnar pillar in the distolabial part of the tooth. The mesialmost lateral pillars are fused to the mesial cingulum, resulting in a complex pattern with two central enamel islands that are fused together and linked by two enamel ridges. There is one median pillar separating each pair of lateral pillars, three in total. The median pillar separating the first and second pair of lateral pillars is cross-shaped and is fused to the second labial lateral pillar. The second median pillar has a more oval shape and is fused to the third labial lateral pillar. The third median pillar is not fused, is slightly smaller, and has a triangular shape. The second pair of lateral pillars have the typical mushroom shape of other *Metridiochoerus* (*Cooke, 1994*), a morphology that is also obvious in the third lingual lateral pillar in this specimen. There is a clear gap between the first and second pair of laterals on the lingual side, which is visibly reduced on the labial side. The length (47.8e mm) and width (16.2 mm) of this specimen are within the range of *M. modestus* from the upper Burgi Member of the Koobi Fora Formation (~2–1.88 Ma), Shungura Member G (~2.27–1.9 Ma), and those from Sterkfontein Member 5 and Bolt's Farm Pit 3 in South Africa (~2–1.7 Ma), but it is shorter than specimens from Konso (~1.9–1.4 Ma), Daka (~1 Ma), and Buia (~1 Ma) (*Cooke, 1993*, *1994*, *2007*; *Gilbert, 2008b*; *Suwa, Souron & Asfaw, 2014*; *Medin et al., 2015*). A.L. 656-1 is narrower than the handful of m3s known for early *M. andrewsi*, such as those from Ledi-Geraru and Mille-Logya (*Lazagabaster et al., 2018*; *Geraads et al., 2021*) and is considerably shorter than those of derived *M. andrewsi*, *M. hopwoodi*, and *M. compactus* from <2 Ma deposits at Koobi Fora and elsewhere (*Harris & White, 1979*).

A.L. 666-16 is a right M2 that is broken both mesially and distally (Figs. 7L and 7M). The tooth is lightly worn, implying that it was erupting at time of death and therefore represents a subadult. The tooth is very narrow at the tip, but the main pillars flare and slope towards the base and give an estimated width of 19.3 mm. The pillars are columnar and intricate, with at least three median pillars in addition to the four main pair of laterals. The distal cingulum is composed of two separated and well-developed pillars.

A.L. 703-1 is a fragment of left mandible with a complete but heavily worn and poorly preserved m2, a small remnant of the distal portion of m1, and the alveolar portion of the roots of p4. Occlusal fusion of the pillars making several enamel islands and the very small size of the alveolus of p4 are typical of *Metridiochoerus*.

A.L. 666-19 is a small mesial fragment of a left m2. It preserves only the metaconid, which has slight wear and a height of approximately 17 mm. The tooth has three small grooves in the superodistal part of the metaconid in lingual view that give this part of the tooth a feather-like appearance, a feature that is characteristic of *Metridiochoerus* but that can also be observed in some specimens of *Kolpochoerus*.

**Discussion:** Though these teeth show some similarities to older samples of *Metridiochoerus andrewsi* (*e.g.*, Ledi-Geraru), the Maka'amitalu sample is a better match for *M. modestus* given the small size of the m3. With the age of the BKT-3 tuff at ~2.35 Ma, these specimens would represent some of the oldest *M. modestus* known in the fossil record, which would explain the primitive aspects of their morphology. *Bishop (2010)* recorded the earliest appearance of *M. modestus* in lower Member G of the Shungura Formation (G-3, ~2.2 Ma), after which it is known from Olduvai Gorge, the Koobi Fora, Nachukui, and Shungura formations, Konso, Buia, and Asbole in eastern Africa.

Order Carnivora Bowdich, 1821
Family Mustelidae Batsch, 1788
Subfamily Lutrinae Bonaparte, 1838
Lutrinae gen. et. sp. indet.

**Referred specimens:** A.L. 1870-1, left mandibular ramus fragment with roots of p3 (distal only), p4, m1, and m2.

**Description:** The ramus is quite broad, but its great height makes it seem slender. Anteriorly, it is broken off between the mesial and distal roots of p3 and posteriorly just after the m2 root. In lateral view the dorsally trending curve of the ramus at the transition from ramus to coronoid process can clearly be seen and shows that the m2, and to some extent the distal part of m1, lie on this upslope. Judging by their roots, p3-m1 were relatively short and broad, and m2 was more or less circular, though possibly slightly broader than long.

**Discussion:** The only carnivoran family that includes members matching the characters of this specimen is Mustelidae. A potential generic assignment is *Enhydriodon* but, as estimated from the roots, the specimen is smaller than any known specimen of *Enhydriodon* younger than ~3 Ma and thus does not match the known record of that taxon. It may potentially belong to the taxon described and illustrated by *Werdelin & Lewis (2013b)*, Fig. 4.16) as Lutrinae gen. et. sp. nov. based on an isolated femur from the upper Burgi Member of the Koobi Fora Formation, but this cannot be verified at present.

Family Hyaenidae Gray, 1821
Hyaenidae gen. et. sp. indet.

**Referred specimens:** A.L. 591-11, right C.

**Description:** Parts of the crown and root are obscured by matrix, but the overall morphology is clear and the specimen can be assigned to Hyaenidae. The specimen is in the size range of *Crocuta* canines, but there are other genera of Hyaenidae in the late Pliocene and Early Pleistocene of Africa that are of similar size (*e.g.*, *Chasmaporthetes*), so the tooth cannot be identified to genus.

Family Felidae Batsch, 1788
Subfamily Felinae Batsch, 1788
Genus *Acinonyx* Brookes, 1828
cf. *Acinonyx* sp.

**Referred specimens:** A.L. 654-1a, head of right metacarpal II.

**Description:** While only the head is preserved, the morphology is clearly felid. The head is relatively narrow mediolaterally and bulbous on the dorsal surface as in the extant cheetah *Acinonyx jubatus*.

Felidae gen. et. sp. indet.

**Referred specimens:** A.L. 654-1b, fifth metacarpal shaft.

**Description:** The proximal end is fragmentary and eroded and the shaft itself is poorly preserved. The shaft is a little curved in comparison to an extant cheetah (*Acinonyx jubatus*) but is consistent with felid morphology.

Subfamily Machairodontinae Gill, 1872
Genus Zdansky, 1924
*Dinofelis* cf. *aronoki* Werdelin & Lewis, 2001

**Referred specimens:** A.L. 1876-1, proximal left humerus.

**Description:** A.L. 1876-1 (Figs. 9A–9D) is nearly identical in size and morphology to KNM-ER 4419, a humerus of the machairodontine *Dinofelis aronoki* from the upper Burgi Member of the Koobi Fora Formation (Werdelin & Lewis, 2001). The head is rounded to a greater degree than in *Homotherium* or larger *Panthera*. While the greater tuberosity projects far enough posteriorly to create a large bicipital groove, it does not project as far as seen in eastern African *Homotherium* (*e.g.*, Werdelin & Lewis, 2013b). The specimen is also much smaller than any known African *Homotherium*.

Order Perissodactyla Owen, 1848
Family Equidae Gray, 1821
Genus *Equus* Linnaeus, 1758
*Equus* sp.

**Referred specimens:** A.L. 587-3, proximal right metatarsal III; A.L. 755-1, ungual phalanx.

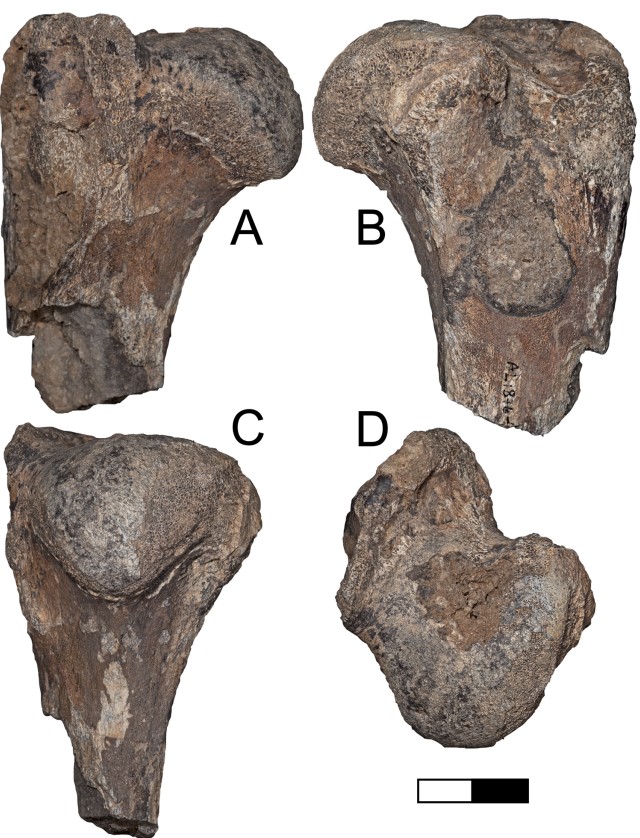

**Figure 9  Maka'amitalu *Dinofelis* cf. *aronoki*.** (A–D) Proximal humerus left A.L. 1876-1 in (A) lateral, (B) medial, (C) posterior, and (D) proximal view. Scale bar equals 2.5 cm.

**Description:** A.L. 587-3 is a proximal metatarsal that resembles *Equus* rather than *Eurygnathohippus* in that the posterior portion of the proximal articular surface is separated from the anterior portion, rather than being connected medially as in *Eurygnathohippus*. A.L. 755-1 is an isolated ungual phalanx smaller than specimens of *Eurygnathohippus* from the temporally older Hadar Formation and late Pliocene Ledi-Geraru sediments. It is a good match in size for the metatarsal, and these presumably belong to the same taxon.

**Discussion:** These two equid postcranial specimens are of similar size and are attributed to the genus *Equus* based on the morphology of the proximal metatarsal A.L. 587-3. Furthermore, measurements of these specimens indicate a species smaller than *Eurygnathohippus* from the Hadar Formation, within the low end of the size range of living *Equus grevyi*, and within the size range of *Eq. oldowayensis* (after *Churcher & Hooijer, 1980*). The proximal anteroposterior and transverse diameters of the Maka'amitalu metatarsal (37.84 × 44.38 mm) are similar to Shungura specimens (*e.g.*, L 7-4 from Member G, 38.5 × 44.5 mm) and fall within the range of those from Olduvai (38.5–47.9 × 43–59.2 mm). The common *Equus* of the eastern African Pleistocene is *Eq. oldowayensis*, which first appears in Member G of the Shungura Formation (~2.27 Ma) and may

have persisted all the way to ~1 Ma based on finds at Daka and Olorgesailie (*Bernor et al., 2010*). While it is possible the Maka'amitalu specimens represent *Eq. oldowayensis*, we refrain from attributing them to a species given the presence of other similarly sized taxa known at this time (*e.g.*, *Eq. koobiforensis*).

Order Primates Linnaeus, 1758
Family Cercopithecidae Gray, 1821
Subfamily Cercopithecinae Gray, 1821
Tribe Papionini Burnett, 1828
Genus *Theropithecus* Geoffroy, 1843
*Theropithecus oswaldi* (Andrews, 1916)
*Theropithecus oswaldi oswaldi* (Andrews, 1916)

**Referred specimens:** A.L. 591-1 (This specimen is listed as A.L. 593-1 in *Frost (2001)*) right mandible with dp3-4; A.L. 592-5 left mandible of a juvenile with a small C fragment, dp3-dp4, m1, and partially erupted m2; A.L. 596-1 mandible of a female with left i1-m2 and right i1-p3; A.L. 653-3 right M, likely M2; A.L. 666-2 distal fragment of right femur; A.L. 666-5 right M3; A.L. 666-7 right I2; A.L. 666-9 right male C; A.L. 666-10 right i; A.L. 666-15 left i2.

**Description:** The preserved dentition is diagnostic for *Theropithecus* showing relatively tall crowns with folded enamel, columnar cusps, deep notches, and flattened basins (*Jolly, 1972*; *Szalay & Delson, 1979*). The enamel folding is fairly well developed (Fig. 10A), which makes the Maka'amitalu *Theropithecus* distinct from ?*T. baringensis* (*Eck & Jablonski, 1987*; *Delson, 1993*; *Gilbert, 2013*). The mandibles, especially A.L. 596-1 (Figs. 10B and 10C), have thick corpora and lack any evidence for corpus fossae, which are typically present in both ?*T. baringensis* and *T. brumpti*. Thus, they preserve the typical morphology of the *Theropithecus oswaldi* lineage and are diagnosably different compared to ?*T. baringensis* and *T. brumpti* (*Eck & Jablonski, 1987*; *Leakey, 1993*; *Delson, 1993*). In size, the dentition is most consistent with *T. o. oswaldi*, but does overlap the range of *T. o. darti* (*Leakey, 1993*; *Frost & Delson, 2002*; *Frost, 2007a*). The incisors (A.L. 666-7, A.L. 666-10, and A.L. 666-15), male canine (A.L. 666-9), and femur (A.L. 666-2, Figs. 10D and 10E) are all only tentatively allocated based on size and a lack of evidence to the contrary, but they could conceivable also represent the same taxon as A.L. 586-10 (see below).
The incisors are all clearly papionin in morphology: the upper has a flaring crown and lacks a lingual cingulum; the lowers lack lingual enamel. The distal femur (A.L. 666-2) is very large and although damaged, appears to show a reverse valgus angle (*Krentz, 1993*; *Gilbert et al., 2011*).

**Discussion:** *Theropithecus oswaldi* is known over a long period of time and is divided into three chronosubspecies: *T. o. darti* (~3.7–2.6 Ma), *T. o. oswaldi* (~2.5–1.6 Ma), and *T. o. leakeyi* (~1.5–0.3 Ma) (*Leakey, 1993*; *Frost & Delson, 2002*; *Frost, 2007a*; *Frost, Jablonski & Haile-Selassie, 2014*). The subspecies are recognized based on a series of clear evolutionary trends that include an increase in cranial, molar, and overall body size,

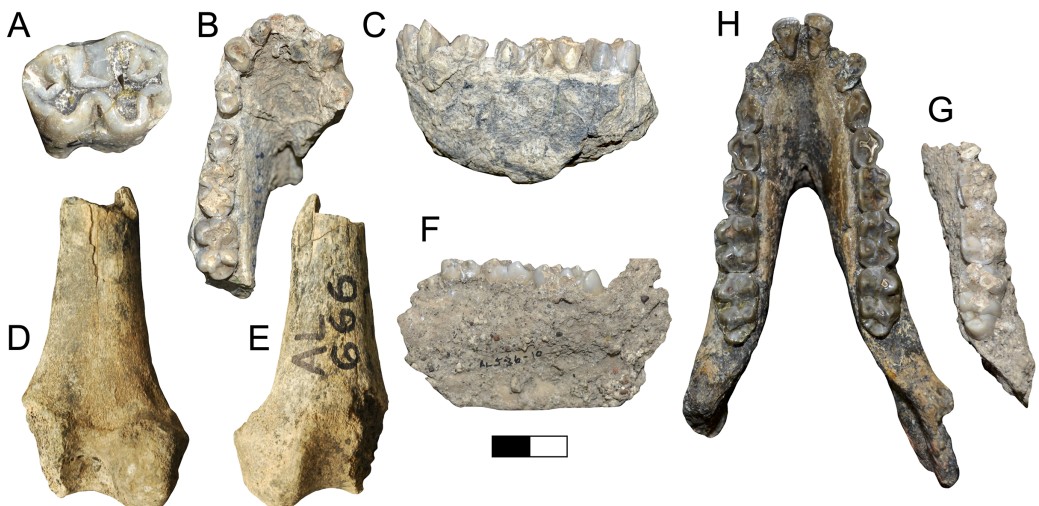

**Figure 10 Maka'amitalu Cercopithecidae.** (A) A.L. 666-5 *Theropithecus oswaldi oswaldi* right M3 in occlusal view; (B and C), A.L. 596-1 *T. o. oswaldi* mandible in occlusal (B) and lateral (C) view; (D and E), A.L. 666-2 *T. o. oswaldi* distal femur in posterior (D) and anterior (E) view. (F and G), A.L. 586-10 Papionini gen. et. sp. indet. mandible in medial (F) and occlusal (G) view, compared to a female specimen (H) of *Soromandrillus quadratirostris* (Omo 47-1970-2008) from the Shungura Formation. Scale bar equals 2 cm.

increasing molar enamel folding and crown height, and cranial adaptations for molar chewing (*Jolly, 1972*; *Leakey, 1993*; *Frost & Delson, 2002*; *Frost, 2007b*). Based on molar size and enamel complexity the Maka'amitalu specimens are too large to be *T. o. darti*, yet are too small and lack the more complexly folded enamel of *T. o. leakeyi*. Their size and enamel complexity are a best overall match for *T. o. oswaldi* (*e.g., Frost, Jablonski & Haile-Selassie, 2014*; *Frost et al., 2017*). Generally, the Maka'amitalu specimens are within the ranges of *T. o. oswaldi* samples from Hata Member and Matabaieutu of the Middle Awash (ca. 2.5 Ma), Shungura Members E through Lower G (2.4–2.1 Ma), and Upper Burgi Member of Koobi Fora (ca. 1.9 Ma) (*Frost, 2001*; *Eck & Jablonski, 1987*; *Jablonski & Leakey, 2008*).

Papionini gen. et sp. indet.

**Referred specimens:** A.L. 586-10, right mandibular *corpus* fragment with m1–m3.

**Description:** This specimen lacks the distinctive molar morphology of *Theropithecus* and is also somewhat smaller in dental and mandibular size than *T. o. oswaldi*. The molars are relatively low-crowned and flaring and otherwise typical of most papionins. In size, they overlap the ranges for *Papio*, *Parapapio*, *Mandrillus*, and *Soromandrillus*, but are slightly smaller than the known range from the relatively small sample of *Gorgopithecus* (*Freedman, 1957*; *Gilbert, Frost & Delson, 2016*; *Gilbert et al., 2018*). The *corpus* is also relatively small and gracile for *Theropithecus oswaldi oswaldi*. With a depth and breadth of approximately 31 mm and 13 mm it is less robust than *T. o. oswaldi* (*Eck & Jablonski, 1987*). The preserved portion of the *corpus* also lacks a *corpus* fossa, which is similar to *Parapapio*, *T. oswaldi* and *S. quadratirostris*, but distinct from *Papio*.

**Discussion:** The molar morphology indicates this specimen represents a taxon other than *Theropithecus* (Figs. 10F and 10G). The lack of mandibular corpus fossae suggests that A.L. 586-10 does not belong to *Papio*. Thus, in terms of size and morphology A.L. 586-10 could potentially represent *Parapapio whitei*, *Soromandrillus quadratirostris*, or *Gorgopithecus major* among known forms. *Parapapio whitei* is currently unknown outside of South Africa, where it has been found from both Makapansgat and Sterkfontein Member 4, ~2.9 to 2 Ma (*Freedman, 1957*; *Delson, 1984*; *Herries et al., 2013*). *Gorgopithecus major* is best known from Kromdraai A, Swartkrans Member 1, and possibly Cooper's D, but has also been recently identified from eastern Africa from Bed I of Olduvai, and is thus generally known from sites younger than 2 Ma (*Freedman, 1957*; *Delson, 1984*; *Gilbert, Frost & Delson, 2016*). *Soromandrillus quadratirostris* is known from Leba, Angola, in southern Africa and the Shungura (Members D through G) and Usno Formations in eastern Africa, and thus spans a considerable time range from 3.4 to 2 Ma (*Delson & Dean, 1993*; *Frost, 2001*; *Gilbert, 2013*). Given the difficulty of diagnosing papionin genera without adequate facial material, we do not make a formal generic allocation here. However, of the known non-*Theropithecus* papionin taxa discussed above, *Soromandrillus* is perhaps the most likely candidate on the basis of geography (eastern Africa) and known chronology (older than ~2 Ma). Omo 47-1970-2008 (Fig. 10H), a mandible of a female *S. quadratirostris* from Member G-8 of the Shungura Formation (~2.1 Ma), is a good morphological match for A.L. 586-10.

Order Proboscidea Illiger, 1811
Family Elephantidae Gray, 1821
Genus *Elephas* Linnaeus, 1758
*Elephas recki* Dietrich, 1915
*Elephas recki atavus* Arambourg, 1947

**Referred specimens:** A.L. 592-2, left m3.

**Description:** A.L. 592-2 is a fragmentary left lower third molar of *Elephas recki* (Figs. 11A and 11B). Comparisons with complete and near complete dental specimens of *E. recki* from the Shungura Formation were used to determine tooth position (T. Getachew, 2020, personal observation). Four lightly worn and a single unworn plate are well preserved, though an unknown number of plates are missing mesially. Some measurements and observations can be made: maximum plate width (81 mm), height (98 mm), general plate length (103+ mm), and lamellar frequency (4.5). The enamel is heavily crenellated and ~3.6 mm thick. The hypsodonty index (height/width $*$ 100) is 129.5, which is within the range of the subspecies *E. r. shungurensis* and *E. r. atavus* (Fig. 11C).

**Discussion:** *Elephas recki* is a long-lived and widely distributed species of the eastern African Plio-Pleistocene. There are several time-successive morphs, recognized as subspecies, are known: *Elephas r. brumpti*, *E. r. shungurensis*, *E. r. atavus*, *E. r. ileretensis*, and *E. r. recki* (*Beden, 1980*). This anagenetically evolving lineage shows progressive increase in the number of tooth plates, enamel folding, and hypsodonty, among other

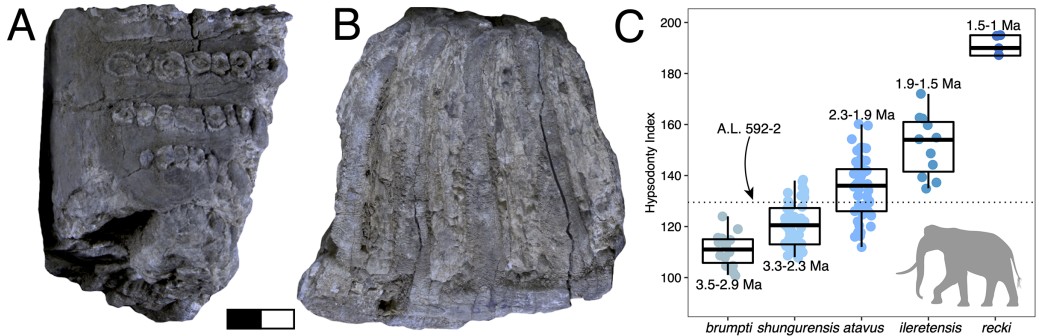

**Figure 11 Maka'amitalu *Elephas recki*.** (A and B), A.L. 592-2 in occlusal (A) and buccal (B) view. (C) Boxplot of hypsodonty indices for the *E. recki* lineage from ~ 3.5–1 Ma for well-constrained sequences in the Turkana Basin (data from *Beden, 1980*, *1983*, *1985*). The hypsodonty index of Maka'amitalu specimen A.L. 592-2 is denoted by the dotted line. Scale bar equals 2 cm.

dental changes, making it an exceptionally good biochronological marker (*Maglio, 1973*; *Beden, 1980*, *1983*; *Sanders et al., 2010*). For example, the oldest subspecies, *E. r. brumpti*, is much less hypsodont (hypsodonty index ~101–116) and has thicker enamel (enamel thickness ~2.8–4 mm) than the terminal subspecies, *E. r. recki* (hypsodonty index ~161–200; enamel thickness ~1.8–3 mm).

Lamellar frequency of the Maka'amitalu specimen (4.5) is similar to the average of *E. r. brumpti*, though there is much overlap between subspecies in this metric and its taxonomic and biochronological utility is therefore questionable. Data from *Beden (1983*, *1985)* show that one can find specimens attributed to nearly every subspecies with lamellar frequency values of 4.5. On the other hand, temporal trends in hypsodonty of the *E. recki* lineage in the well-constrained strata of the Turkana Basin indicate that this trait is a robust biochronological indicator (Fig. 11C). The hypsodonty index of A.L. 592-2 (129.5) falls within the ranges of *E. recki shungurensis* (~3.3–2.3 Ma) and *E. recki atavus* (~2.3–1.9 Ma) but fits more comfortably in the latter. We therefore attribute the Maka'amitalu m3 to *E. recki atavus*.

## Turnover and paleoecology

The Maka'amitalu fauna shares few species with the older faunas from Ledi-Geraru and the Hadar Formation. These include *Sivatherium maurusium*, *Kolpochoerus phillipi*, the *Elephas recki* and *Theropithecus oswaldi* lineages, and probably also *Kobus sigmoidalis* and *Eudorcas praethomsoni*. Among herbivores (Artiodactyla, Perissodactyla, Proboscidea), $\beta_{sim}$ values for adjacent submembers of the Hadar Formation are low ($\beta_{sim}$ range = 0.08–0.24) but gradually increase through time (Fig. 12A) indicating slightly greater turnover towards the top of the formation. In contrast, turnover was very high between the top of the Hadar Formation (KH-2) and the Gurumaha fault block of Ledi-Geraru ($\beta_{sim}$ = 0.53) and again between the Lee Adoyta fault block and the Maka'amitalu ($\beta_{sim}$ = 0.58). Closer inspection reveals that $\beta_{sim}$ values are strongly predicted by the difference in midpoint age (*i.e.*, mean of the upper and lower bounding ages) between adjacent fossil units (OLS regression, r2 = 0.83). Indeed, pairwise comparisons of $\beta_{sim}$

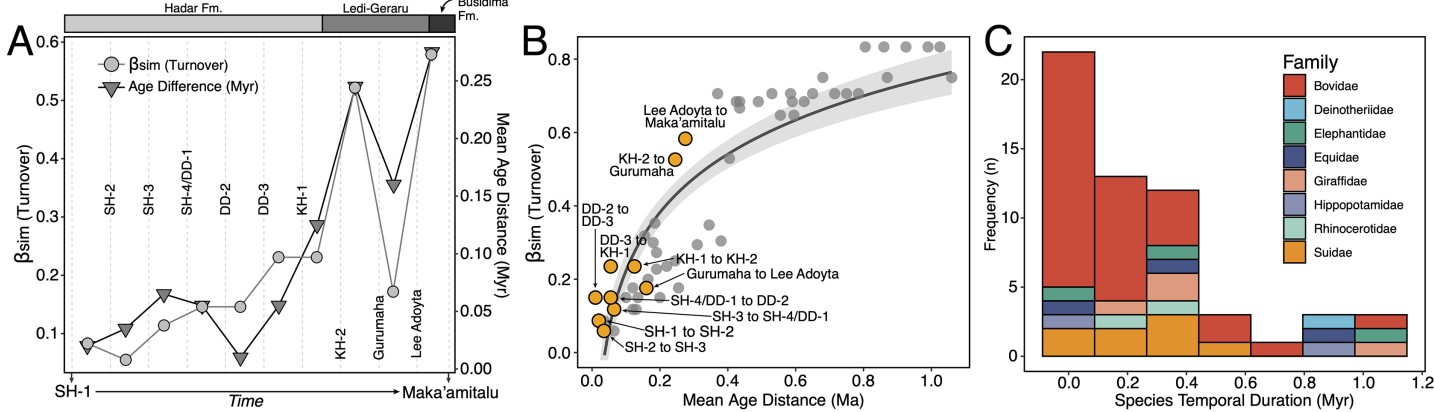

**Figure 12 Taxonomic turnover in the lower Awash Valley between ~3.45–2.35 Ma.** (A) Simpson dissimilarity ($\beta_{sim}$) and mean age distances (Myr) across adjacent fossil units in the lower Awash sequence. (B) Scatterplot of pairwise $\beta_{sim}$ values against their log mean age distance, with adjacent fossil units (bin-to-bin comparisons) in the sequence (*e.g.*, KH-1 to KH-2) shown by orange points and all other pairwise comparisons shown by gray points. (C) Species temporal durations (*i.e.*, time from first appearance datum to the last appearance datum) in the lower Awash sequence, showing the lognormal distribution of species temporal durations.

values *vs* their log mean age distance across all units in the lower Awash sequence shows that taxonomic turnover increases relatively quickly in relation to age distance, but then gradually flattens towards peak similarity (Fig. 12B). One would expect that the magnitude of taxonomic turnover between fossil units should strongly correlate with their age difference if origination, extinction, and dispersal were occurring continuously through time, and if species temporal duration frequencies were lognormally distributed (Fig. 12C). This is the case for the lower Awash sequence, with no clear deviations from the general pattern (*i.e.*, no instances where species turnover is far greater than would be expected by age distance). Thus, the apparent 'pulses' of turnover in the lower Awash sequence at ~2.9–2.8 Ma and again at ~2.5–2.3 Ma are most parsimoniously explained by temporal discontinuities in the fossil record, which would suggest that species replacement occurred at a constant rate through time in the lower Awash Valley between ~3.45–2.35 Ma.

Functional ecological turnover, as recorded by herbivore diet and body size proportions, follows a similar pattern (Figs. 13A and 13B). The diet structure of the Maka'amitalu herbivore assemblage is dominated by grazing taxa, with less common browsing and mixed-feeding taxa occurring in similar proportions. Overall, browser proportions were fairly constant through the lower Awash sequence, with the decline of mixed-feeders and rise of grazers between the KH-2 submember of the Hadar Formation and Gurumaha fault block of Ledi-Geraru driving the largest shift. Among body size classes, there appears to have been a loss of large-bodied (s6, >750 kg) taxa alongside gains of smaller-bodied (s1–s3, <18–150 kg) species through time. However, this probably reflects changes in depositional environments and sample sizes rather than a change in ecological structure. For example, hippopotamids are quite common in the fluvio-lacustrine deposits of the Hadar Formation (*n* > 200 specimens collected; Campisano et al., in press) but are represented by only two specimens, likely from the same individual, from the

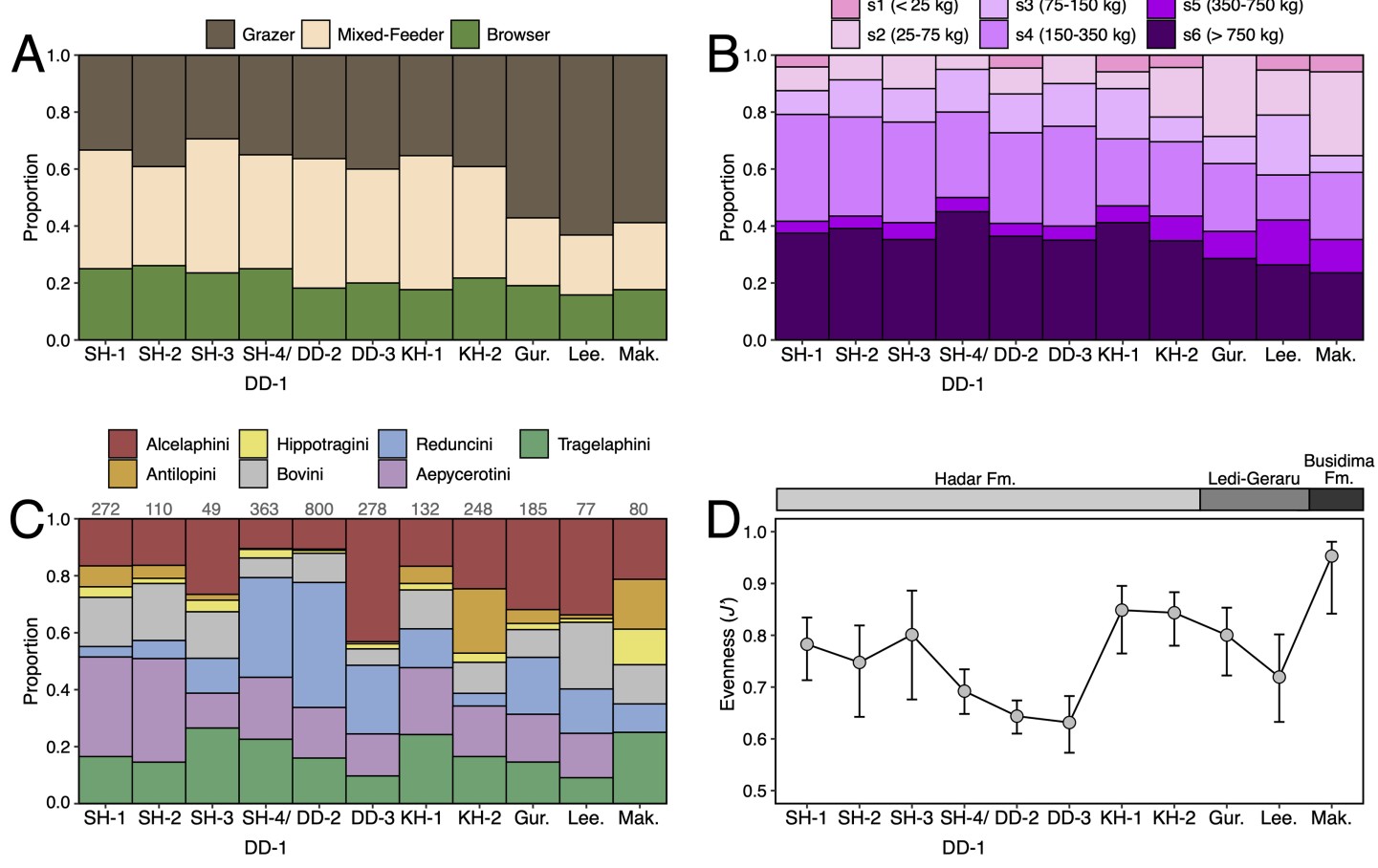

**Figure 13 Ecological turnover in the lower Awash Valley between ~3.45–2.35 Ma.** (A) Dietary proportions. (B) Body size class proportions. (C) Relative abundance of bovid tribes, with values on top of each bar representing the bovid NISP for the assemblage. (D) Evenness of bovid tribal abundances measured by the Pielou index ($J'$) with error bars representing bootstrapped 95% confidence intervals.

higher-energy fluvial deposits of Maka'amitalu. Given the presence of other large-bodied taxa (*e.g.*, deinotheres and rhinocerotids) at non-Maka'amitalu Busidima Formation localities in the Hadar Research Project area and nearby Early Pleistocene sites in the Afar (*e.g.*, Uraitele exposures of Mille-Logya; *Geraads et al., 2021*), it seems likely that the rarity of large-bodied herbivores in the Maka'amitalu assemblage reflects depositional and/or taphonomic differences and not ecology.

Changes in the relative abundance of bovid tribes throughout the lower Awash sequence are shown in Fig. 13C. When plotted by submember, no clear trend emerges through the Hadar Formation; the few notable changes, such as the abundance of reduncins in DD-2 and antilopins in KH-2, are discussed by *Reed (2008)* and *Geraads, Bobe & Reed (2012)*. Relative abundances for Maka'amitalu are as follows: Tragelaphini (25%), Alcelaphini (21.3%), Antilopini (17.5%), Bovini (13.8%), Hippotragini (12.5%), Reduncini (10%). Given the small sample size (NISP, $n = 80$), it seems sensible to consider these abundances as roughly equivalent. Evenness of tribal abundances (Fig. 13D) is highest for the Maka'amitalu assemblage ($J' = 0.95$). Indeed, evenness values are negatively correlated

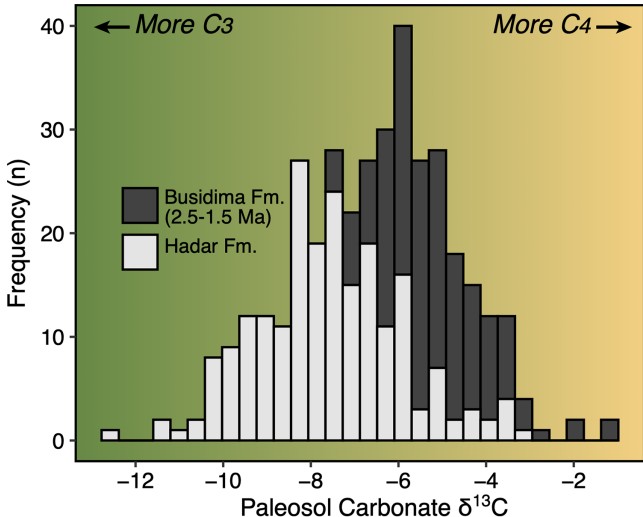

**Figure 14 Paleovegetation change across the Hadar and Busidima formations as recorded by stable carbon isotopes of paleosol carbonates.** Data from *Levin et al. (2004)*, *Quade et al. (2004)*, *Aronson, Hailemichael & Savin (2008)*, and *Cerling et al. (2011)*. 

with bovid NISP through the lower Awash sequence (r = −0.58), which probably suggests that smaller assemblages are strongly influenced by taphonomic and/or sampling variation and that more specimens are required to detect an ecological signal. Perhaps a notable exception is the absence of impala (Aepycerotini) from the Maka'amitalu sample despite their commonness (~20–30% of all bovid craniodental remains) across the submembers of the Hadar Formation and, to a lesser extent, Ledi-Geraru (~15–20%). This might suggest a further reduction in wooded or ecotonal habitats in Maka'amitalu times, which is consistent with the preponderance of grazers among the herbivore fauna. Previously published stable carbon isotopes from Busidima Formation paleosols in the Dikika, Gona, and Hadar areas demonstrate significantly greater $C_4$ biomass in the lower Awash Valley from ~2.5–1.5 Ma compared to the environments recorded by the Hadar Formation (Fig. 14). Finally, although not included in our analysis of the lower Awash Valley herbivore faunas, *Theropithecus* is not rare (*n* = 10) in the Maka'amitalu assemblage. Data from *Wynn et al. (2016)* show that Maka'amitalu teeth of this taxon clearly belonged to $C_4$-dominated consumers (mean $\delta^{13}C$ = −1 ± 0.7‰ *n* = 3).

## Biochronological implications

The first appearance datum (FAD), last appearance datum (LAD), and distribution of mammal taxa recorded as present, or likely present (either as 'cf.' taxa or chronospecies lineages), in the Maka'amitalu assemblage and from other sites in eastern Africa dating to ~2.5–1 Ma are given in Table 2. In terms of chronological ranges, most taxa are consistent with the age of the BKT-3 tuff (2.35 ± 0.07 Ma), with the records of *Elephas recki atavus* and *Equus* approximating their FADs in the upper Member F to Member G interval (~2.3 Ma) of the Shungura Formation (*Bernor et al., 2010*; *Sanders et al., 2010*). Based on a linear model of the areas of all six molar positions (upper and lower M1-M3) against geological age (r2 = 0.83), the molar size of the Maka'amitalu *Theropithecus oswaldi*

Peer J

**Table 2  Distribution of taxa present (or likely present) in the Maka'amitalu assemblage from other sites in eastern Africa dating ~2.5–1 Ma, with regional first (FAD) and last (LAD) appearance data.**

| Family | Species | FAD | LAD | Ref. | Hata 2.50 | Mb D 2.49 | Uraitele 2.46 | Lokalalei 2.43 | Mb E 2.36 | Mb F 2.30 | Kalochoro 2.10 | Mb G 2.07 | Upper Burgi 1.94 | Bed I 1.92 | Konso 1 1.85 | Mb H 1.82 | Lower Bed II 1.77 | Konso 2 1.75 | Middle Bed II 1.72 | Kaitio 1.71 | KBS 1.70 | Mb J 1.65 | Konso 3 1.55 | Upper Bed II 1.47 | Mb K 1.46 | Okote 1.46 | Boolihanan 1.26 |
|---|---|---|---|---|---|---|---|---|---|---|---|---|---|---|---|---|---|---|---|---|---|---|---|---|---|---|---|
| Bovidae | *Beatragus antiquus* | 2.50 | 1.46 | a | | | | | | | | 1 | 1 | 1 | | | | | | 1 | | | | | | 1 | |
| Bovidae | *Eudorcas praethomsoni* | 3.52 | 0.01 | a | | | | | 1 | 1 | | 1 | | | | 1 | | | | 1 | | | | | 1 | 1 | |
| Bovidae | *Kobus sigmoidalis-ellipsiprymnus* | 3.80 | extant | a | 1 | 1 | 1 | | 1 | 1 | 1 | 1 | 1 | 1 | 1 | | 1 | 1 | 1 | 1 | 1 | 1 | 1 | 1 | 1 | | 1 |
| Bovidae | *Parmularius altidens-angusticornis* | 2.11 | 0.70 | a | | | | | | | | 1 | | 1 | | 1 | | 1 | | 1 | | | | 1 | | | |
| Bovidae | *Tragelaphus strepsiceros* | 2.50 | extant | b | 1 | | | | | | 1 | 1 | 1 | 1 | | | 1 | | | 1 | 1 | | | 1 | | 1 | 1 |
| Cercopithecidae | *Theropithecus oswaldi oswaldi* | 2.50 | 1.50 | c | 1 | | | | 1 | 1 | 1 | 1 | 1 | 1 | 1 | | 1 | | | 1 | 1 | 1 | 1 | | | | 1 |
| Elephantidae | *Elephas recki atavus* | 2.34 | 1.50 | d | | | | | 1 | 1 | 1 | 1 | 1 | 1 | 1 | 1 | 1 | 1 | | | | | 1 | | | | |
| Equidae | *Equus* spp. | 2.27 | 1.00 | e | | | | | | | | 1 | 1 | 1 | 1 | 1 | 1 | 1 | 1 | 1 | 1 | 1 | 1 | 1 | 1 | 1 | 1 |
| Felidae | *Dinofelis aronoki* | 3.50 | 1.70 | f | | | | | | | | | 1 | | | | | | | | 1 | | | | | | |
| Giraffidae | *Sivatherium maurusium* | 4.10 | 0.50 | g h | | | 1 | 1 | | 1 | 1 | 1 | 1 | 1 | | | 1 | | | 1 | | | | 1 | 1 | 1 | 1 |
| Hippopotamidae | *Hippopotamus gorgops* | 2.50 | 0.60 | i | | | | | | | | | 1 | 1 | | | 1 | | | 1 | 1 | | | 1 | | 1 | |
| Suidae | *Kolpochoerus phillipi-majus* | 2.80 | 0.10 | j k | | | | | | | | | | | 1 | | | 1 | | | | | 1 | | | | |
| Suidae | *Metridiochoerus modestus* | 2.21 | 0.70 | l | | | | | | | | | 1 | 1 | | | 1 | | | 1 | 1 | | 1 | 1 | | 1 | 1 |

**Notes:**

Sites are listed chronologically from left to right by their mean age. Presence-absence data from *de Heinzelin et al. (1999)*; *Suwa et al. (2003)*; *Everett (2010)*; *Kovarovic, Slepkov & McNulty (2013)*; *Werdelin & Lewis (2013b)*; *Fortelius et al. (2016)*; *Bibi et al. (2018)*; *Du & Alemseged (2018)*; *Geraads et al. (2021)*.
[a] *Bibi & Kiessling (2015)*.
[b] *Bibi (2009)*.
[c] *Jablonksi & Frost (2010)*.
[d] *Sanders et al. (2010)*.
[e] *Bernor et al. (2010)*.
[f] *Werdelin & Lewis (2005)*.
[g] *Harrison (2011)*.
[h] *Harris, Solounias & Geraad (2010)*.
[i] *Weston & Boisserie (2010)*.
[j] *Lazagabaster et al. (2018)*.
[k] *Souron, Boisserie & White (2015)*.
[l] *Bishop (2010)*.

*oswaldi* specimens has a best fit age of approximately 2.3 Ma with a 95% confidence range of 2.7–1.9 Ma (Frost et al., in prep.). Similarly, the molars of Maka'amitalu *Kolpochoerus* specimens present an intermediate morphology between *Kolpochoerus phillipi* from Ledi-Geraru and Matabaietu (~2.8–2.5 Ma) and *K.* cf. *majus* from Konso (~1.9–1.4 Ma). Finally, another probable transitional form comes from the *Syncerus* remains, which differ in horn core morphology and dental size in ways that align them with *Gentry's (1985) Syncerus* '?*acoelotus*' sample from Shungura Formation Members C and G when compared to the Olduvai type series of *S. acoelotus* (*Gentry & Gentry, 1978*).

On the other hand, some mammal taxa suggest a slightly younger age. The Maka'amitalu *Parmularius* horn seems to represent the *Parmularius altidens-angusticornis* lineage best known from Olduvai Beds I-II (~1.9–1.2 Ma), but probably also present in Shungura Member G (G7 and G10-11, ~2.1 Ma) and Member H (1.87–1.76 Ma). Earlier specimens that *Gentry (2011)* aligned with this lineage as *P.* ?*altidens* from the upper Ndolanya Beds of Laetoli tend to be quite small (see Fig. 4 B). The only other option for a *Parmularius* this large >2 Ma would be *P. pachyceras*, which occurs at both Hadar and Ledi-Geraru (*Bibi, Rowan & Reed, 2017*), but to which the Maka'amitalu specimen clearly does not belong. Likewise, the Maka'amitalu hippopotamid postcrania indicates the presence of a species similar in size to those that appear in the Turkana Basin ~2 Ma at the transition between the lower and upper parts of Member G in the Shungura Formation and in the upper Burgi Member of the Koobi Fora Formation (Figs. 6F and 6G). The earliest appearance of *M. modestus* comes from lower Member G (G-3, ~2.2 Ma) of the Shungura Formation. At ~2.35 Ma, the Maka'amitalu *Metridiochoerus modestus* would push back the FAD of this species, though the morphology of the molars is not dissimilar to those of *M. modestus* samples ~2 Ma and younger.

In sum, the taxonomic composition and morphological stages of the Maka'amitalu fauna suggests that an age range of 2.3–1.9 Ma is most likely for the assemblage. Obtaining a refined biochronological estimate is compromised by the scarcity of 2.5–2 Ma faunas from the Afar. *Geraads et al. (2021)* recently described the mammalian fauna recovered from the Uraitele exposures (2.5–2.4 Ma) of Mille-Logya, just northeast of the Hadar area, but few species are shared with the Maka'amitalu. A similar pattern emerges from comparisons with the ~2.5 Ma Hata, Gamedah, and Matabaietu faunas from the Middle Awash, though only a handful of taxa have been fully described from these assemblages (*e.g.*, *Vrba, 1997*). Outside of the Afar, it is difficult to know how biochronological comparisons may be influenced by geographic, environmental, or depositional differences. Undoubtedly the best record of 2.5–2 Ma faunas in eastern African comes from the Shungura Formation (*Boisserie et al., 2008*), which crops out in the lower Omo Valley ~900 km south of the Hadar area. The Shungura record, however, comprises fluvial deposits associated with the ancestral Omo River and is well-known to sample much woodier habitats compared to other sites (*Levin et al., 2011*).

## CONCLUSIONS

Early Pleistocene faunas from the Afar Depression are poorly known compared to the region's rich Pliocene deposits. Though small and fragmentary, the Maka'amitalu large

mammal fauna provides an important datapoint in this regard. Combined with the older Ledi-Geraru and Hadar Formation faunas, the Maka'amitalu assemblage caps a >1-Myr record of hominin and mammalian evolution in the lower Awash Valley. The fauna recovered from the Maka'amitalu basin comprises at least 20 large mammal taxa, all of which are broadly consistent with an Early Pleistocene age. Together, the age of BKT-3 and biochronology suggest an age range of 2.4–1.9 Ma for the faunal assemblage. Species turnover across the Hadar-Ledi-Geraru-Maka'amitalu sequence is highly correlated with the age distance between adjacent fossil units, which could suggest a constant rate of origination, extinction, and/or dispersal through time. Functional ecological turnover follows a similar pattern of gradual change through the sequence, though some changes (*e.g.*, the decline of large-bodied taxa) likely reflect changes in depositional settings, taphonomy, and sample sizes rather than ecosystem structure. Continued work in the Maka'amitalu basin by the Hadar Research Project and by teams in adjacent project areas, such as Ledi-Geraru and Mille-Logya, promises to continue to improve our knowledge of the Early Pleistocene period in the lower Awash Valley and, more broadly, the Afar Depression.

## ACKNOWLEDGEMENTS

We thank the Authority for Research and Conservation of Cultural Heritage (ARCCH), Ethiopian Ministry of Culture and Tourism, and the Afar Regional State for permission to conduct field work at Hadar. Thanks to the ARCCH staff at the National Museum of Ethiopia for curation of the Hadar Research Project collections and for facilitating our study of the Maka'amitalu fauna. We thank D. Geraads and one anonymous reviewer for their valuable suggestions, as well as editor M. Sponheimer.

### Funding

Research funding for the Hadar Research Project was provided by the Institute of Human Origins and the National Science Foundation. Ignacio A Lazagabaster was supported by a postdoctoral fellowship from the Humboldt Foundation. Faysal Bibi was supported by Deutsche Forschungsgemeinschaft (DFG) project number 282995372 (BI 1879/1-1). The funders had no role in study design, data collection and analysis, decision to publish, or preparation of the manuscript.

### Grant Disclosures

The following grant information was disclosed by the authors:
Institute of Human Origins and the National Science Foundation.
Humboldt Foundation.
Deutsche Forschungsgemeinschaft (DFG): 282995372 (BI 1879/1-1).

### Competing Interests

The authors declare that they have no competing interests.

John Rowan
Ignacio A. Lazagabaster
Christopher J. Campisano
Faysal Bibi
René Bobe
Jean-Renaud Boisserie
Stephen R. Frost
Tomas Getachew
Christopher C. Gilbert
Margaret E. Lewis
Sahleselasie Melaku
Eric Scott
Antoine Souron
Lars Werdelin
William H. Kimbel
Kaye E. Reed
2022
10.7717/peerj.13210

## Author Contributions

- John Rowan conceived and designed the experiments, performed the experiments, analyzed the data, prepared figures and/or tables, authored or reviewed drafts of the paper, and approved the final draft.
- Ignacio A. Lazagabaster conceived and designed the experiments, performed the experiments, analyzed the data, prepared figures and/or tables, authored or reviewed drafts of the paper, and approved the final draft.
- Christopher J. Campisano conceived and designed the experiments, performed the experiments, analyzed the data, authored or reviewed drafts of the paper, and approved the final draft.
- Faysal Bibi performed the experiments, analyzed the data, authored or reviewed drafts of the paper, and approved the final draft.
- René Bobe performed the experiments, analyzed the data, authored or reviewed drafts of the paper, and approved the final draft.
- Jean-Renaud Boisserie performed the experiments, analyzed the data, authored or reviewed drafts of the paper, and approved the final draft.
- Stephen R. Frost performed the experiments, analyzed the data, authored or reviewed drafts of the paper, and approved the final draft.
- Tomas Getachew performed the experiments, analyzed the data, authored or reviewed drafts of the paper, and approved the final draft.
- Christopher C. Gilbert performed the experiments, analyzed the data, authored or reviewed drafts of the paper, and approved the final draft.
- Margaret E. Lewis performed the experiments, analyzed the data, authored or reviewed drafts of the paper, and approved the final draft.
- Sahleselasie Melaku performed the experiments, analyzed the data, authored or reviewed drafts of the paper, and approved the final draft.
- Eric Scott performed the experiments, analyzed the data, authored or reviewed drafts of the paper, and approved the final draft.
- Antoine Souron performed the experiments, analyzed the data, authored or reviewed drafts of the paper, and approved the final draft.
- Lars Werdelin performed the experiments, analyzed the data, authored or reviewed drafts of the paper, and approved the final draft.
- William H. Kimbel conceived and designed the experiments, performed the experiments, analyzed the data, authored or reviewed drafts of the paper, and approved the final draft.
- Kaye E. Reed conceived and designed the experiments, performed the experiments, analyzed the data, authored or reviewed drafts of the paper, and approved the final draft.

### Field Study Permissions

The following information was supplied relating to field study approvals (*i.e.*, approving body and any reference numbers):

The Authority for Research and Conservation of Cultural Heritage (ARCCH), Ethiopian Ministry of Culture and Tourism, and the Afar Regional State granted permission to conduct field work at Hadar.

### Data Availability

All raw measurements and datasets analyzed for paleoecological and paleobiological analyses (functional trait, turnover, and relative abundances for lower Awash taxa) are available in the Supplemental File.

### Supplemental Information

Supplemental information for this article can be found online at http://dx.doi.org/10.7717/peerj.13210#supplemental-information.

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
