# Peer review of "Early Pleistocene large mammals from Maka’amitalu, Hadar, lower Awash Valley, Ethiopia"

_PeerJ, doi:10.7717/peerj.13210_

## Round 0.1 · original submission · Minor Revisions

This looks to be in very good shape. Please address reviewer comments as appropriate. Where you do not address reviewer concerns simply explain your reasoning in an accompanying letter.

Reviewer 1 ·

Basic reporting

I’d like to commend the authors for providing an overall well-written, grounded and sober analysis of the Maka’amitalu faunas. Relative to some trends in the recent literature, I appreciate the efforts made by the authors to provide primary descriptions and grounded systematic palaeontology within the context of a peer-reviewed submission instead of an emphasis on offering just a ‘species table’ or sole emphasis on interpretation over primary data on the fossil materials.
Most of my comments are brief and, in most cases, merely reflect stylistic considerations that the authors can elect to accept/reject. A few more substantive comments do crop up, and I hope the authors will consider expanding a bit of the manuscript to address them.
Minor Comments
Lines 57-59: The authors use terms like ‘medium’, ‘large’ and ‘megaherbivore’ within the abstract that don’t really come up much within the remainder of the document (megaherbivore, for example) and aren’t grounded by a definition as to what designates these categories. Body size is integrated into the faunal analyses but follows a more discrete system. I appreciate the authors are trying to delineate their consideration of ‘macromammals’ vs. ‘micromammals’, but absent some sort of criterion I just assume… the taxa described are the ones collected and there isn’t a collection of collected but undescribed materials?

Line 78: ‘witnessed’ is a bit awkward
Line 84: ‘formations’ vs. ‘Formations’ – given that later and in most broad use a singular Formation (e.g., Shungura Formation) is capitalised, would capitalising this be appropriate? The use here is in a list of named ‘formations’ rather than use of the term generally.
Lines 100-104: Might consider reordering the sentence structure. I don’t know that the taxonomic turnover and palaeoecological analyses of faunal change come ‘from’ the systematic description as currently phrased.

Line 145 (and across manuscript): ‘orders’ vs. ‘Orders’ as a formal Linnean rank
Line 286: ‘Younger in age is Syncerus acoelotus…’ is a bit of an awkward word ordering to my ear.
Line 308 (and across manuscript): The use of ‘pedicel’ vs. ‘pedicle’ – I’m just flagging that the use of the former in the manuscript somewhat falls in the less common occurrence in the literature than the latter.
Line 395: Overall the authors are great with their anatomical nomenclature, and I appreciate the challenges of describing horn core shapes/orientations; ‘backwards’ somewhat stands out as a result, but may be unavoidable.
Line 656: In this circumstance I appreciate the authors are wanting to signal a potential allocation here, though equally one could argue that the ambiguity in the ability to attribute these specimens to either genus could push towards ‘Reduncini gen. et sp. indet’ with notes on the potential allocation.

Line 690: Given the defined use of dental nomenclature, ‘lower p4’ seems redundant (and only seems to arise here).

Lines 712-759: I appreciate the challenges of wrangling multiple, taxon-focused contributors into a single voice within a manuscript. That said, the section on the hippo remains stands out as being quite stylistically different and with a number of awkward sentence structures/grammatical choices. I would suggest a hand in editing to eliminate some unnecessary commas (across section) and run on sentences (e.g., 729-733).

Experimental design

Geological Setting Section (pre- Line 135): I would appreciate some bridging paragraph or sentence(s) to clarify whether all the ~40m of Busidima formations are exposed within the limited Maka’amitalu area that was sampled. The overall description of the geological setting is comprehensive, just lost within this is whether all these described tuffs and sequences are potentially represented within the sampling area. This speaks to an additional question, which is whether the authors could expand a bit on the nature of the potential time averaging and the internal integrity of the ‘assemblage’. When moving forward with ecological and turnover analyses there is the underlying assumption of association, but how reasonable that assumption is comes from how confident the associations are between samples (or at least tempered with known sources of distortion in the sample). As the authors note, it’s a relatively small faunal sample to start with and from surface exposures… so a bit of unpacking around what makes this a discrete sample to analyse would help with later discussion/interpretations (e.g., abundance data and cross-site comparisons in lines 1155-onwards).

Validity of the findings

Lines 870-911: Given where the identification of four specimens lands (particularly with biochronological implications that are later discussed in lines 1178-on), I will admit to being let down by the discussion section that simple states that the Maka’amitalu specimens ‘is a better match for M. modestus’ in lieu of a clear basis for attributing the materials as such relative to other metriodiochoerines (e.g., the noted similarities to M. andrewsi). This does not mean that the authors are ultimately incorrect in the attribution, but the entire primary description provided here is more focused on a clear case for the generic attribution… but no specific call outs to provide strong support for the ultimate specific diagnosis. With this attribution later noted as a potential extension of the FAD, I really urge the authors to clarify within the description/discussion of these specimens why they are M. modestus to the exclusion of M. andrewsi beyond the relatively vague statement that it is “similar to M. andrewsi… but we are going with M. modestus”. This is really my only substantive concern in the primary attribution of the remains across the entire manuscript.

Turnover and Paleoecology/Geological Setting (Lines 1114-on): On reaching this section and considering the results of the analysis, I would suggest a bit more information (if possible) on the nature of the depositional settings and whether there is supporting information on the nature of the hydrological setting for the sampled region? In line 1150 we get the statement that there are ‘higher-energy fluvial deposits’ at Maka’amitalu, but does that mean that the entire sequence is fluvial? All high energy? I don’t see this content in the prior geological setting description, and a somewhat clearer description of what is known about the hydrological setting across the set of deposits potentially aggregated in this assemblage would assist a bit in the context of interpreting the local regional palaeoecology.

·

Basic reporting

This is an excellent contribution to the knowledge of Eastern African hominid environments, written by several of the best specialists of fossil mammals of this period. The presentation and style are clear, the introduction and 'Geological setting' clearly present the context, the length of the paper is neither too short nor too long, the figures are good (minor corrections needed -see below), the raw data are provided (except perhaps for Theropithecus – see below). The paper should definitely be published, with only a few minor changes or improvements.

Experimental design

All requirements (primary research within Aims and scope, research question ell defined, relevant and meaningful, rigorous investigation, details and information provided) are fulfilled, but see 3. Validity of the findings

Validity of the findings

My main concern is about chronology. The authors conclude to an age range of 2.4 – 1.9 Ma but there are several indications that the lower part of this range is unlikely; I suspect that this extension of the range towards early ages has been influenced by the dating of the BKT-3, but this is irrelevant to biochronology.
My arguments are:
- first, the presence of Equus unambiguously points to an age younger than 2.3 Ma
- the authors themselves believe that Hippopotamus cf. gorgops points to an age not older 2 Ma
- the authors' estimate of a gap of almost 0.6 Ma (their Fig. 12A) between Lee Adoyta and Maka'amitalu also suggest that their own age estimate is close to 2 Ma…
- l. 1043-1044, we are told that the Maka'amitalu specimens 'fit best with T. o. oswaldi', and l. 1187, it is mentioned that the 'best fit' based on T. oswaldi is 2.4 Ma; I am sure that C. Gilbert and S. Frost have good arguments for that, but they should be provided.
- l. 1213 – 1214, the authors correctly observe that 'few species [from the Uraitele exposures] are shared with the Maka'amitalu'. But, since the sites are geographically close, isn't this precisely because the ages are different ? The differences in faunal composition with the nearby MLP sites (even accounting for the possible Kolpochoerus/Metridiochoerus issue) is striking, and points to a significant age difference.
In my opinion, the authors should either provide biochronological arguments for extending the possible time-range towards older dates, or address the disagreement between the radiometric age and the biochronological data.

Additional comments

Additional, minor comments:
l. 84: capitalize Formations ?
check lines 112-113
l. 168-170: some bovids are small and very likely to be impacted by taphonomic and collection biases
l. 437-438: Beatragus is also known from Ahl al Oughlam, Morocco
l. 450: what is the meaning of 'Parmularius spp. (cf. Damaliscus spp.)' ?
l. 519: 'that is very slender'
l. 521: backward
l. 576 et seq.: on the basis of the photos, it is difficult for me to imagine that this is right horncore, so Parantidorcas can probably be excluded from the comparison.
l. 983: assignment to E. cf. oldowayensis is poorly supported; either provide stronger arguments, or just call it Equus sp.
l. 988-989: I would use 'anterior' and 'posterior' rather than 'dorsal' and 'ventral'
l. 1000: 'L 7-4 from member G'
l. 1083: explain in a few words why this is an m3; please note that Beden (1987, tab. 42) mentions an m3 of E.recki ileretensis whose hypsodonty index is 116. Couldn't the hypsodonty index of your tooth be higher than 129.5 in another, missing part of the crown ? I would not identify your tooth fragment to subspecies.
l. 1154: don't you know what were the sampling procedures ? Since members of the Hadar Research Project are among the authors, they should know whether the hippos were systematically collected or not.
Figures: several of the symbols of XY plots are quite hard to see. Please check.
Fig. 2: since the caption mentions "Hadar Research Project area" it is awkward to plot here localities that are no longer part of this Research Project; either remove them or add a historical note.
Fig. 5. check captions for LMNO (JKLM in the figure)
Fig 7: scales bars are 40 and 10 mm (not cm)
Fig. 12C: this is a personal opinion, but I think this is highly sensitive to incomplete sampling

Reference list
Jablonski & Frost 2010, Vrba 1995 are not mentioned in the text
Szalay & Delson 1979, Gentry 2006 are missing in the ref. list
Boisserie & Gilbert 2008 or 2009 ?
Souron et al. 2013 or 2015 ?
Bengtson, not Bengston
Delson et al., not Delson & Dean
Locke & Rowan 2016, not Locke et al.
Frost 2007a & b in the ref. list
Give full ref. for Geraads et al. 2021

---

## Round 0.2 · accepted · Accept

Very nice letter addressing reviewer concerns--a good faith effort was made to address these and changes were made where warranted. "We have kept the original wording but thank the reviewer for this suggestion" was no doubt appropriate on several occasions.